# Fast to Train, Fast to Sample: Stable Velocity for Flow Matching

## Abstract

We revisit flow matching from a variance-centric perspective. Although conditional flow matching (CFM) is theoretically elegant, its use of single-sample conditional velocities introduces high variance, which can destabilize optimization and slow convergence. We demonstrate that this behavior induces two distinct regimes: a high-variance regime that hinders training and a low-variance regime where conditional and true velocities are nearly identical, thereby enabling analytical sampling shortcuts. Motivated by these insights, we introduce the **Stable Velocity** framework to improve both the training and sampling processes of flow matching. For training, we propose *Stable Velocity Matching (StableVM)*, a variance-reduced objective that preserves CFM's global optima while significantly improving stability and convergence in the high-variance regime. For sampling, we introduce *Stable Velocity Sampling (StableVS)*, a "free lunch" acceleration method that leverages the low-variance regime to achieve faster generation without requiring finetuning. Experiments on SiT-XL trained on ImageNet, as well as on several large pretrained models (SD3, SD3.5, Flux, and Wan2.2), show consistent improvements in training convergence and more than $2\times$ faster sampling while maintaining high fidelity.

## 1 Introduction

Recent years have seen significant progress in generative modeling with the advent of diffusion (Sohl-Dickstein et al., 2015; Ho et al., 2020; Song et al., 2020; Karras et al., 2022b), flow matching (Lipman et al., 2022; Liu et al., 2022), and stochastic interpolants (Albergo et al., 2023; Ma et al., 2024; Yu et al., 2024). These approaches model the transformation of a simple prior distribution, *e.g.*, Gaussian noise, into a complex data distribution by a time-dependent stochastic or deterministic process. Through this lens, diffusion and flow-based models can be unified as different instantiations of stochastic interpolants, providing a principled foundation for large-scale generative modeling. Their success has fueled breakthroughs across diverse applications such as high-fidelity image synthesis (Labs, 2024; Esser et al., 2024), image restoration (Rombach et al., 2022; Lin et al., 2024), and video generation (Brooks et al., 2024; Wan et al., 2025).

Among these advances, the Conditional Flow Matching (CFM) framework (Tong et al., 2023) has emerged as an elegant formulation. Instead of explicitly simulating the forward stochastic differential equation (SDE) or probability flow ordinary differential equation (PF-ODE), CFM directly trains a neural network to predict the conditional velocity field that transports intermediate states along the interpolant. At convergence, the learned neural velocity field provably coincides with the true probability flow (Tong et al., 2023; Vincent, 2011; Xu et al., 2023), offering both theoretical soundness and practical scalability. Consequently, CFM has become a cornerstone objective for training flow and diffusion-based generative models.

Despite its theoretical elegance, CFM has a critical, yet often overlooked, weakness: the variance of its training target. In practice, the conditional velocity $v_t(x_t \mid x_0)$ used in CFM is only a single-sample Monte Carlo estimate of the true velocity $v_t(x_t)$. This introduces potentially high variance, particularly at timesteps where the marginal distribution remains close to the simple prior. High-variance targets can destabilize

optimization, slow convergence, and create a mismatch between the empirical training dynamics and the ideal global optimum. While empirical studies have repeatedly hinted at such variance-induced inefficiencies in diffusion training (Karras et al., 2022a; Choi et al., 2022; Xu et al., 2023), a general variance-theoretic understanding within flow matching and broader stochastic interpolant frameworks has been lacking.

In this work, we develop a *variance-based perspective* on stochastic interpolants. By explicitly quantifying the variance of conditional velocities, we uncover a two-regime structure that governs training and inference: 1) The *high-variance regime* arises at timesteps closer to the prior (high-noise), where conditional velocity deviates substantially from the true velocity field. 2) The *low-variance regime* arises at timesteps closer to the data distribution (low-noise), where the conditional velocity aligns closely with the true velocity field. Building on these observations, we propose **Stable Velocity**, a novel framework for both training and sampling in flow matching. In particular, for training, we propose **Stable Velocity Matching (StableVM)**, which provably shares the same global optima as CFM but reduces variance by aggregating multiple reference samples. For sampling, our variance analysis reveals that once variance is negligible, the dynamics become stable and admit closed-form simplifications that reduce the number of steps required for inference without sacrificing sample quality. We call this "free-lunch" acceleration **Stable Velocity Sampling (StableVS)**. We validate our approach across SiT-XL trained on ImageNet, and state-of-the-art pretrained generative models. StableVM consistently improves training convergence (Fig. 3 and 5), while StableVS accelerates inference by more than $2\times$ (*e.g.*, 50 steps to 25 steps) for the latest flow-based models, including Stable Diffusion 3, Stable Diffusion 3.5, Flux, and Wan2.2, without quality degradation. Our main contributions are summarized as follows:

- **Variance analysis of stochastic interpolants.** We provide the first systematic study of variance on stochastic interpolants, identifying a two-regime structure that governs training and inference.

- **Stable Velocity Matching (StableVM).** We propose an unbiased, variance-reduced training objective that provably retains the same global optima of CFM while dramatically lowering training variance.

- **Stable Velocity Sampling (StableVS).** We exploit the low-variance regime to derive simplified dynamics that enable accelerated and stable sampling in pretrained models without additional finetuning.

## 2 PRELIMINARIES

We present a brief overview of flow matching and stochastic interpolants. Their detailed connections to score-based diffusion models are provided in Appendix A.

**Flow Matching and Stochastic Interpolants.** Given samples from an unknown data distribution $q(\boldsymbol{x}_0)$ over $\mathbb{R}^d$, the goal of generative modeling is to learn a model capable of generating new samples from $q$. Flow matching (Lipman et al., 2022; Liu et al., 2022; Lipman et al., 2024b) and stochastic interpolants (Albergo et al., 2023; Albergo & Vanden-Eijnden, 2023) approach this task by defining a continuous-time stochastic process. Starting with a data point $\boldsymbol{x}_0 \sim q$ and Gaussian noise $\boldsymbol{\varepsilon} \sim \mathcal{N}(0, \boldsymbol{I})$, the interpolant is defined as

$$\boldsymbol{x}_t = \alpha_t \boldsymbol{x}_0 + \sigma_t \boldsymbol{\varepsilon}, \quad \alpha_0 = \sigma_1 = 1, \; \alpha_1 = \sigma_0 = 0, \tag{1}$$

where $\alpha_t$ and $\sigma_t$ are differentiable functions satisfying $\alpha_t^2 + \sigma_t^2 > 0$ for all $t \in [0, 1]$. In practice, most works (Ma et al., 2024; Yu et al., 2024; Lipman et al., 2022; Liu et al., 2022; Lipman et al., 2024b) adopt a simple linear interpolant $\alpha_t = 1-t$, $\sigma_t = t$. This process induces a conditional and a marginal velocity field,

$$\boldsymbol{v}_t(\boldsymbol{x}_t \mid \boldsymbol{x}_0) = \frac{\sigma_t'}{\sigma_t}(\boldsymbol{x}_t - \alpha_t \boldsymbol{x}_0) + \alpha_t' \boldsymbol{x}_0 \quad (2) \qquad \boldsymbol{v}_t(\boldsymbol{x}_t) = \mathbb{E}_{p_t(\boldsymbol{x}_0 \mid \boldsymbol{x}_t)}\left[\boldsymbol{v}_t(\boldsymbol{x}_t \mid \boldsymbol{x}_0)\right] \quad (3)$$

such that the probability flow ordinary differential equation (PF-ODE)

$$\mathrm{d}\boldsymbol{x}_t = \boldsymbol{v}_t(\boldsymbol{x}_t)\, \mathrm{d}t \tag{4}$$

induces marginal distributions that match that of Eq. (1) for all time $t \in [0, 1]$. Sampling from the model thus reduces to solving the PF-ODE in Eq. (4) using standard ODE solvers, *e.g.*, Euler integration, starting from Gaussian noise $\boldsymbol{\varepsilon} \sim \mathcal{N}(0, \boldsymbol{I})$ (Lipman et al., 2022; Ma et al., 2024).

In addition, there exists a reverse stochastic differential equation (SDE) whose marginal $p_t(x)$ coincides with that of the PF-ODE in Eq. (4), but with an added diffusion term (Ma et al., 2024):

$$\mathrm{d}\boldsymbol{x}_t = \boldsymbol{v}_t(\boldsymbol{x}_t)\,\mathrm{d}t - \tfrac{1}{2}w_t\boldsymbol{s}_t(\boldsymbol{x}_t)\,\mathrm{d}t + \sqrt{w_t}\,\mathrm{d}\overline{\mathbf{w}}_t, \tag{5}$$

where $\overline{\mathbf{w}}_t$ is a standard Wiener process in backward time, $\sqrt{w_t}$ is the diffusion coefficient, and the score $\boldsymbol{s}_t(\boldsymbol{x}_t) = \nabla_{\boldsymbol{x}_t}\log p_t(\boldsymbol{x}_t)$ can be re-expressed using the velocity field (Ma et al., 2024):

$$\boldsymbol{s}_t(\boldsymbol{x}_t) = \sigma_t^{-1}(\alpha_t\boldsymbol{v}_t(\boldsymbol{x}_t) - \alpha_t'\boldsymbol{x}_t)/(\alpha_t'\sigma_t - \alpha_t\sigma_t'). \tag{6}$$

**Conditional Flow Matching (CFM).** Training typically uses the CFM objective (Tong et al., 2023):

$$\min_{\boldsymbol{\theta}} \quad \mathbb{E}_{t,\,q(\boldsymbol{x}_0),\,p_t(\boldsymbol{x}_t|\boldsymbol{x}_0)}\lambda_t\left\|\boldsymbol{v}_{\boldsymbol{\theta}}(\boldsymbol{x}_t,t) - \boldsymbol{v}_t(\boldsymbol{x}_t\mid\boldsymbol{x}_0)\right\|^2, \tag{7}$$

where $\lambda_t$ is a positive weighting function, and $\boldsymbol{v}_{\boldsymbol{\theta}}(\cdot,\cdot) : [0,1]\times\mathbb{R}^d\to\mathbb{R}^d$ is a neural velocity field parameterized by $\boldsymbol{\theta}$. Here, the conditional path distribution is $p_t(\boldsymbol{x}_t\mid\boldsymbol{x}_0) = \mathcal{N}(\boldsymbol{x}_t\mid\alpha_t\boldsymbol{x}_0, \sigma_t^2\boldsymbol{I})$. The minimizer of Eq. (7) is provably the true marginal velocity field (Tong et al., 2023; Xu et al., 2023; Ma et al., 2024):

$$\boldsymbol{v}_{\boldsymbol{\theta}}^*(\boldsymbol{x}_t,t) = \mathbb{E}_{p_t(\boldsymbol{x}_0|\boldsymbol{x}_t)}\left[\boldsymbol{v}_t(\boldsymbol{x}_t\mid\boldsymbol{x}_0)\right] = \boldsymbol{v}_t(\boldsymbol{x}_t). \tag{8}$$

**Variance of CFM.** Although CFM provides an unbiased estimator of the true velocity field $\boldsymbol{v}_t(\boldsymbol{x}_t)$, it suffers from a key limitation: its training target, $\boldsymbol{v}_t(\boldsymbol{x}_t\mid\boldsymbol{x}_0)$, is only a single-sample Monte Carlo estimate of the expectation in Eq. (3). This introduces high variance (Owen, 2013; Elvira & Martino, 2021), which can slow convergence and degrade performance. Following Xu et al. (2023), we quantify this variance by the average trace of the conditional velocity variance at time $t$:

$$\begin{aligned}\mathcal{V}_{\mathrm{CFM}}(t) &= \mathbb{E}_{p_t(\boldsymbol{x}_t)}\left[\mathrm{Tr}\left(\mathrm{Cov}_{p_t(\boldsymbol{x}_0|\boldsymbol{x}_t)}(\boldsymbol{v}_t(\boldsymbol{x}_t\mid\boldsymbol{x}_0))\right)\right]\\ &= \mathbb{E}_{q(\boldsymbol{x}_0),\,p_t(\boldsymbol{x}_t|\boldsymbol{x}_0)}\left[\|\boldsymbol{v}_t(\boldsymbol{x}_t\mid\boldsymbol{x}_0) - \boldsymbol{v}_t(\boldsymbol{x}_t)\|^2\right].\end{aligned} \tag{9}$$

This definition mirrors the CFM loss in Eq. (7) without the expectation over $t$ and with the neural velocity $\boldsymbol{v}_{\boldsymbol{\theta}}$ replaced by the true velocity $\boldsymbol{v}_t(\boldsymbol{x}_t)$. Intuitively, a small $\mathcal{V}_{\mathrm{CFM}}(t)$ means that $\boldsymbol{v}_t(\boldsymbol{x}_t\mid\boldsymbol{x}_0)$ is close to the true velocity $\boldsymbol{v}_t(\boldsymbol{x}_t)$, whereas a large value indicates strong deviations.

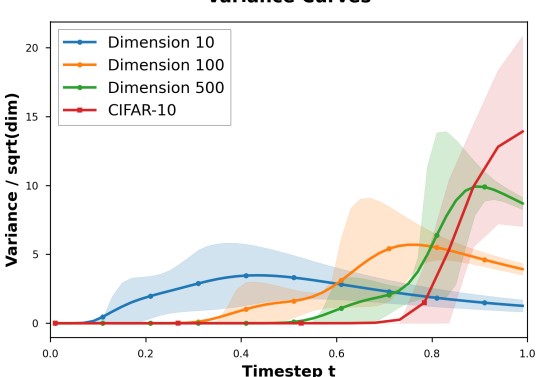

Figure 1: Variance curves of $\mathcal{V}_{\mathrm{CFM}}(t)$ with 15%–85% quantile intervals, evaluated on GMMs with varying dimensionality and on CIFAR-10. The $y$-axis shows $\mathcal{V}_{\mathrm{CFM}}(t)$ normalized by the square root of data dimension. More details are provided in Appendix E.1.4.

To better understand the behavior of $\mathcal{V}_{\mathrm{CFM}}(t)$, we evaluate it on Gaussian Mixture Models (GMMs) and CIFAR-10 (Krizhevsky, 2009). As shown in Fig. 1, we make two consistent observations:

1. For all cases, the variance remains close to zero at small $t$ but increases rapidly as $t$ grows. This naturally divides the process into two regimes separated by a split point $\xi$: a *low-variance regime* ($0 \le t < \xi$) with $\mathcal{V}_{\mathrm{CFM}}(t) \approx 0$, and a *high-variance regime* ($\xi \le t \le 1$) where $\mathcal{V}_{\mathrm{CFM}}(t)$ is significantly larger.

2. As the data dimensionality grows, the split point $\xi$ shifts closer to 1, enlarging *low-variance regime* while compressing *high-variance regime*. At the same time, the overall variance magnitude increases.

Fig. 2 further illustrates these phenomena. In the *low-variance regime*, $\boldsymbol{x}_t$ is often influenced by a single sample, so the conditional velocity nearly coincides with the true velocity, keeping $\mathcal{V}_{\mathrm{CFM}}(t)$ small. In the

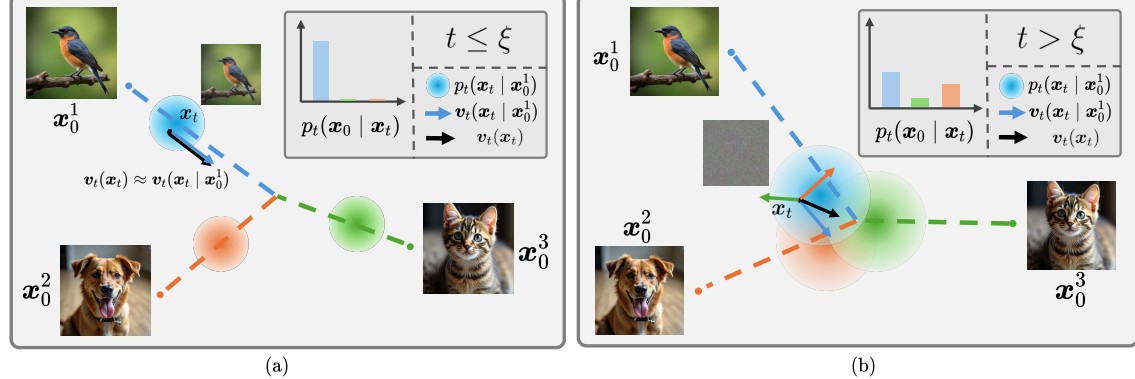

(a)                                    (b)

Figure 2: Illustration of CFM variance $\mathcal{V}_{\text{CFM}}(t)$. (a) The *low-variance regime* ($t \leq \xi$), where the posterior $p_t(\boldsymbol{x}_0 \mid \boldsymbol{x}_t)$ is sharply concentrated and the conditional velocity $\boldsymbol{v}_t(\boldsymbol{x}_t \mid \boldsymbol{x}_0)$ nearly coincides with the true velocity $\boldsymbol{v}_t(\boldsymbol{x}_t)$, yielding $\mathcal{V}_{\text{CFM}}(t) \approx 0$. (b) The *high-variance regime* ($t > \xi$), the posterior spreads over multiple reference samples, causing the conditional velocity to fluctuate and resulting in a large $\mathcal{V}_{\text{CFM}}(t)$.

*high-variance regime*, $\boldsymbol{x}_t \sim p_t$ is influenced by multiple samples, causing the conditional velocity to deviate substantially from the true velocity and leading to higher variances. As dimensionality increases, the distances between data points grow, delaying this mixing effect and shifting the split point $\xi$ closer to 1.

These observations naturally suggest two key research directions: 1) In the *high-variance regime*, can we design methods to reduce variance to improve training convergence while preserving the correct global minimizer? 2) In the *low-variance regime*, where the velocity field already aligns closely with the true velocity field, can we develop sampling strategies that speed up generation without compromising performance?

## 3 VARIANCE-DRIVEN OPTIMIZATION OF TRAINING AND SAMPLING

In this section, we address the two key questions raised in Sec. 2 by introducing **Stable Velocity Matching (StableVM)**, a new variance-reduced training objective for stochastic interpolants, and **Stable Velocity Sampling (StableVS)**, an accelerated sampling method that does not require finetuning.

### 3.1 STABLE VELOCITY MATCHING: REDUCING TRAINING VARIANCE

**Stable Velocity Matching (StableVM).** We first introduce our novel training objective—StableVM and then show that this objective shares exactly the same global minimizer as CFM in Eq. (8). Furthermore, we quantify how its trace-of-variance decays as the reference batch size grows. Inspired by Xu et al. (2023), we introduce $n$ reference samples $\{\boldsymbol{x}_0^i\}_{i=1}^n$, drawn i.i.d. from the data distribution $q(\boldsymbol{x}_0)$. We then define a composite conditional probability path $p_t^{\text{GMM}}\left(\boldsymbol{x}_t \mid \{\boldsymbol{x}_0^i\}_{i=1}^n\right) := \sum_{i=1}^n \frac{1}{n} p_t(\boldsymbol{x}_t \mid \boldsymbol{x}_0^i)$, which is essentially a mixture of conditional probabilities associated with each reference sample. We can derive the corresponding posterior distribution as in the following proposition. The proof is in Appendix D.1.

**Proposition 1.** *The posterior* $p_t^{\text{GMM}}\left(\{\boldsymbol{x}_0^i\}_{i=1}^n \mid \boldsymbol{x}_t\right) = \frac{1}{n} \sum_{i=1}^n \left(p_t\left(\boldsymbol{x}_0^i \mid \boldsymbol{x}_t\right) \prod_{j \neq i} q(\boldsymbol{x}_0^j)\right).$

Based on the reference samples and $p_t^{\text{GMM}}$, we formulate our proposed unbiased training objective within the general stochastic interpolant framework. This objective function, denoted as $\mathcal{L}_{\text{StableVM}}$, is defined as:

$$
\begin{aligned}
\mathcal{L}_{\text{StableVM}}(\boldsymbol{\theta}, t) &= \mathbb{E}_{\{\boldsymbol{x}_0^i\}_{i=1}^n \sim q^n, \, \boldsymbol{x}_t \sim p_t^{\text{GMM}}(\cdot \mid \{\boldsymbol{x}_0^i\}_{i=1}^n)} \left\| \boldsymbol{v}_{\boldsymbol{\theta}}(\boldsymbol{x}_t, t) - \sum_{k=1}^n \frac{p_t(\boldsymbol{x}_t \mid \boldsymbol{x}_0^k) \boldsymbol{v}_t(\boldsymbol{x}_t \mid \boldsymbol{x}_0^k)}{\sum_{j=1}^n p_t(\boldsymbol{x}_t \mid \boldsymbol{x}_0^j)} \right\|^2 \\
&= \mathbb{E}_{\boldsymbol{x}_t \sim p_t, \, \{\boldsymbol{x}_0^i\}_{i=1}^n \sim p_t^{\text{GMM}}(\cdot \mid \boldsymbol{x}_t)} \left\| \boldsymbol{v}_{\boldsymbol{\theta}}(\boldsymbol{x}_t, t) - \sum_{k=1}^n \frac{p_t(\boldsymbol{x}_t \mid \boldsymbol{x}_0^k) \boldsymbol{v}_t(\boldsymbol{x}_t \mid \boldsymbol{x}_0^k)}{\sum_{j=1}^n p_t(\boldsymbol{x}_t \mid \boldsymbol{x}_0^j)} \right\|^2.
\end{aligned}
\tag{10}
$$

Our StableVM target is compatible with general stochastic interpolant framework. Under the special case of VP diffusion, it closely resembles the STF objective Eq. (20), but differs in the construction of the composite conditional probability path $p_t^{\text{GMM}}$. A detailed discussion is provided in Appendix B. As shown in the following theorem, the use of $p_t^{\text{GMM}}$ is the essential ingredient that guarantees the unbiasedness of our target. The detailed proof is provided in Appendix D.2.

**Theorem 1.** *(a) The StableVM target is unbiased. That is, for any $\boldsymbol{x}_t$, we have*

$$\mathbb{E}_{\{\boldsymbol{x}_0^i\}_{i=1}^n \sim p_t^{GMM}(\cdot|\boldsymbol{x}_t)} \left[ \sum_{k=1}^n \frac{p_t(\boldsymbol{x}_t \mid \boldsymbol{x}_0^k)\boldsymbol{v}_t(\boldsymbol{x}_t \mid \boldsymbol{x}_0^k)}{\sum_{j=1}^n p_t(\boldsymbol{x}_t \mid \boldsymbol{x}_0^j)} \right] = \boldsymbol{v}_t(\boldsymbol{x}_t).$$

*(b) The global minimizer $\boldsymbol{v}^*(\boldsymbol{x}_t, t)$ of the StableVM objective $\mathcal{L}_{StableVM}$ is the true velocity field $\boldsymbol{v}_t(\boldsymbol{x}_t)$.*

**Variance of StableVM.** Having established its unbiasedness, we now analyze the variance of StableVM, highlighting its significantly reduced training target variance compared to CFM. To quantify this, we follow Eq. (9) and consider the average trace-of-variance for StableVM:

$$
\begin{aligned}
\mathcal{V}_{\text{StableVM}}(t) &= \mathbb{E}_{\boldsymbol{x}_t \sim p_t} \left[ \text{Tr} \left( \text{Cov}_{\{\boldsymbol{x}_0^i\}_{i=1}^n \sim p_t^{\text{GMM}}(\cdot|\boldsymbol{x}_t)} \left( \sum_{k=1}^n \frac{p_t(\boldsymbol{x}_t \mid \boldsymbol{x}_0^k)\boldsymbol{v}_t(\boldsymbol{x}_t \mid \boldsymbol{x}_0^k)}{\sum_{j=1}^n p_t(\boldsymbol{x}_t \mid \boldsymbol{x}_0^j)} \right) \right) \right] \\
&= \mathbb{E}_{\boldsymbol{x}_t \sim p_t, \{\boldsymbol{x}_0^i\}_{i=1}^n \sim p_t^{\text{GMM}}(\cdot|\boldsymbol{x}_t)} \left\| \boldsymbol{v}_t(\boldsymbol{x}_t) - \sum_{k=1}^n \frac{p_t(\boldsymbol{x}_t \mid \boldsymbol{x}_0^k)\boldsymbol{v}_t(\boldsymbol{x}_t \mid \boldsymbol{x}_0^k)}{\sum_{j=1}^n p_t(\boldsymbol{x}_t \mid \boldsymbol{x}_0^j)} \right\|^2
\end{aligned}
\tag{11}
$$

These equivalent expressions provide several ways to represent the variance of the StableVM training target, characterizing the mean squared error between the estimated target and the true velocity $\boldsymbol{v}_t(\boldsymbol{x}_t)$.

In the following theorems, we compare $\mathcal{V}_{\text{StableVM}}$ against $\mathcal{V}_{\text{CFM}}$. The proofs are provided in Appendix D.3. We first show that $\mathcal{V}_{\text{StableVM}}$ is always upper bounded by $\mathcal{V}_{\text{CFM}}$.

**Theorem 2.** *Fix $t \in [0, 1]$. We always have $\mathcal{V}_{StableVM}(t) \leq \mathcal{V}_{CFM}(t)$.*

We then provide a stronger variance bound in the large sample regime. In particular, the upper bound gets stricter with a factor of $O(1/n)$.

**Theorem 3.** *Fix $t \in [0, 1]$. Let $\boldsymbol{v}_t$ be bounded. Let $\varepsilon \in (0, 1)$. Assume*

$$M := \int_{\{\boldsymbol{x}:p_t(\boldsymbol{x})\leq\varepsilon\}} \mathbb{E}_{\boldsymbol{x}_0 \sim p_t(\cdot|\boldsymbol{x}_t)} \left[ \|\boldsymbol{v}_t(\boldsymbol{x}_t \mid \boldsymbol{x}_0) - \boldsymbol{v}_t(\boldsymbol{x}_t)\|^2 \right] \, \mathrm{d}\boldsymbol{x}_t < \infty.$$

*Then, for large enough $n$, we have*

$$\mathcal{V}_{StableVM}(t) \lessapprox \frac{1}{n-1} \left( \frac{1}{\varepsilon} \cdot \mathcal{V}_{CFM}(t) + M \right).$$

We note that the (informal) assumption of $n$ being large enough comes from the limits in distribution of the random variables as $n \to \infty$ as in Lemmas 3 and 4. We also note that we can choose a suitable $\varepsilon$ in practice such that the set $\{\boldsymbol{x} : p_t(\boldsymbol{x}) \leq \varepsilon\}$ is "small" (in the sense that it has Lebesgue measure close to 0), which would lead to $M$ being close to 0.

**Extension to Class-Conditional Generation with Classifier-Free Guidance.** Thus far, we have discussed variance reduction in the unconditional setting. Extending StableVM to conditional generation poses additional challenges. The main difficulty lies in the sparsity of conditional distributions: if we directly adopt the on-the-fly sampling strategy from Alg. 1, only a few reference samples will match a given label or prompt, leading to an insufficiently small effective reference set. To address this, we introduce a memory bank mechanism tailored for class-conditional generation. The memory bank is pre-populated with reference samples from the entire training dataset prior to training. During training, this memory bank is updated using a First-In, First-Out (FIFO) policy. This design guarantees that each class maintains a sufficiently large and diverse pool of reference samples, thereby stabilizing the training objective. The full procedure is detailed in Alg. 2.

---

**Algorithm 1** Stable Velocity Matching

---

**Require:** Training iteration $T$, initial model $v_\theta$, dataset $\mathcal{D}$, learning rate $\eta$, ema rate $\alpha$.
1: **for** $iter = 1 \ldots T$ **do**
2:     Sample a batch $\mathcal{B} = \{x_0^i\}$ from $\mathcal{D}$, $|\mathcal{B}| > 1$
3:     Uniformly sample time $t \sim q_t(t)$ from $[0, 1]$
4:     Sample perturbed batch $\{x_t^j\}_{j=1}^M$ from $p_{\text{GMM}}(x_t^j \mid \{x_0^i\}) = \sum_{x_0 \in \mathcal{B}} \frac{1}{|\mathcal{B}|} p_t(x_t^j \mid x_0)$
5:     Calculate stable vector field for all $x_t^j$: $v_\mathcal{B}(x_t^j) = \sum_{x_0 \in \mathcal{B}} \frac{p_t(x_t^j \mid x_0)}{\sum_{y_0 \in \mathcal{B}} p_t(x_t^j \mid y_0)} \cdot v_t(x_t^j \mid x_0)$
6:     Calculate loss for $v_\theta$: $\mathcal{L}(\theta) = \frac{1}{M} \sum_{j=1}^M \lambda(t) \| v_\theta(x_t^j, t) - v_\mathcal{B}(x_t^j) \|^2$
7:     Update model $v_\theta$: $\theta = \theta - \eta \nabla \mathcal{L}(\theta)$
8: **end for**
9: **return** $v_\theta$

---

## 3.2 STABLE VELOCITY SAMPLING: ACCELERATING SAMPLING AT LOW-VARIANCE REGIME

As discussed in Sec. 2 and shown in Fig 2, the variance $\mathcal{V}_{\text{CFM}}(t)$ is nearly zero in the *low-variance regime*. In this case, the true velocity $v_t(x_t)$ is effectively determined by a single data point $x_0$ and can be well approximated by the conditional velocity field $v_t(x_t \mid x_0)$. In other words, once the dominant data point $x_0$ is identified from $x_t$ and $v_t(x_t)$, the analytical form of the velocity field—and hence its trajectory—becomes available. This observation enables stable and analytical simulation of the reverse dynamics between timesteps $\tau$ and $t$ such that $\tau < t \leq \xi$. Specifically, we substitute the true velocity field $v_t(x_t)$ with its conditional counterpart $v_t(x_t \mid x_0)$ in the PF-ODE (Eq. (4)) and the reverse SDE (Eq. (5)), and then perform the corresponding integration. If the trained model $v_\theta(x_t, t)$ perfectly recovers $v_t(x_t)$, this yields a principled way to accelerate sampling without sacrificing performance.

**StableVS for SDE.** For the reverse SDE (Eq. (5)), this yields the following DDIM-style (Song et al., 2021) posterior distribution:

$$p_\tau(x_\tau \mid x_t, v_t(x_t)) = \mathcal{N}\left( \mu_{\tau|t}, \ \beta_t^2 \mathbf{I} \right), \tag{12}$$

where $\beta_t = f_\beta \sigma_\tau$, the factor $f_\beta \in [0, 1]$, and the posterior mean is

$$\mu_{\tau|t} = \left( \sqrt{\frac{\sigma_\tau^2 - \beta_t^2}{\sigma_t^2}} - \left( \alpha_\tau - \alpha_t \sqrt{\frac{\sigma_\tau^2 - \beta_t^2}{\sigma_t^2}} \right) \cdot \frac{\sigma_t'/\sigma_t}{\alpha_t' - \alpha_t \sigma_t'/\sigma_t} \right) x_t + \left( \alpha_\tau - \alpha_t \sqrt{\frac{\sigma_\tau^2 - \beta_t^2}{\sigma_t^2}} \right) \cdot \frac{v_t(x_t)}{\alpha_t' - \alpha_t \sigma_t'/\sigma_t}, \tag{13}$$

where $\alpha_t'$ and $\sigma_t'$ represent the time derivatives of $\alpha_t$ and $\sigma_t$. The derivation is provided in Appendix D.4.

**StableVS for ODE.** For the probability flow ODE, the exact solution at timestep $\tau$ is

$$x_\tau = \sigma_\tau \left[ \left( \frac{1}{\sigma_t} - \frac{\sigma_t'}{\sigma_t} \cdot \frac{I(t, \tau)}{C_t} \right) x_t + \frac{I(t, \tau)}{C_t} v_t(x_t) \right], \tag{14}$$

where $C_t := \alpha_t' - \alpha_t \sigma_t'/\sigma_t$, $I(t, \tau) := \int_t^\tau C(s)/\sigma_s ds$. The derivation is in Appendix D.5. In the special case of linear interpolant (*i.e.*, $\alpha_t = 1 - t$, $\sigma_t = t$), setting $\beta_t = 0$ in Eq. (12) makes two samplers coincide:

$$x_\tau = x_t + (\tau - t) v_t(x_t). \tag{15}$$

For this interpolant, Eq. (15) shows that the PF-ODE trajectory reduces to a straight line with constant velocity $v_t(x_t)$, allowing exact integration via Euler steps of arbitrary size within the *low-variance regime*.

Table 1: ImageNet $256 \times 256$ evaluation with different bank sizes under StableVM. All metrics are measured with the SDE Euler-Maruyama sampler (NFE=250) with classifier-free guidance. ↓ and ↑ indicate whether lower or higher values are better, respectively.

| Iterations | Model | FID↓ | sFID↓ | IS↑ | Precision↑ | Recall↑ |
|---|---|---|---|---|---|---|
| 250k | SiT-XL/2 | 6.71 | 4.76 | 147.8 | 0.78 | 0.53 |
| | + StableVM (bank=256) | 6.50 | 4.74 | 154.7 | 0.78 | 0.53 |
| | + StableVM (bank=512) | 6.35 | 4.61 | 152.8 | 0.79 | 0.53 |
| | + StableVM (bank=1024) | **5.85** | **4.57** | **157.4** | 0.79 | 0.53 |
| 500k | SiT-XL/2 | 4.01 | 4.53 | 188.1 | 0.80 | 0.55 |
| | + StableVM (bank=256) | 3.66 | 4.52 | 197.0 | 0.81 | 0.55 |
| | + StableVM (bank=512) | 3.39 | 4.46 | 204.1 | 0.81 | 0.56 |
| | + StableVM (bank=1024) | **3.20** | **4.46** | **208.9** | 0.81 | 0.56 |
| 750k | SiT-XL/2 | 2.84 | 4.42 | 219.7 | 0.81 | 0.57 |
| | + StableVM (bank=256) | 2.79 | 4.46 | 220.9 | 0.81 | 0.56 |
| | + StableVM (bank=512) | 2.75 | 4.46 | 225.0 | 0.82 | 0.57 |
| | + StableVM (bank=1024) | **2.50** | 4.53 | **238.1** | 0.82 | 0.56 |
| 1M | SiT-XL/2 | 2.41 | 4.43 | 236.2 | 0.82 | 0.58 |
| | + StableVM (bank=256) | **2.38** | 4.57 | **242.4** | 0.82 | 0.57 |
| | + StableVM (bank=512) | 2.39 | **4.41** | 238.5 | 0.82 | 0.57 |
| | + StableVM (bank=1024) | 2.39 | 4.46 | 237.1 | 0.82 | **0.58** |

## 4 EXPERIMENTS

In this section, we empirically validate our Stable Velocity framework through extensive experiments. In Section 4.1, we demonstrate the effectiveness of our StableVM target by training an SiT-XL (Ma et al., 2024) on ImageNet (Deng et al., 2009). We also conduct a two-stage training and sampling experiment that exploits the high and low variance regimes. In Section 4.2, we verify our StableVS sampling acceleration on large-scale pretrained text-to-image models. Further details on experiments are provided in Appendix E.

### 4.1 EVALUATION ON STABLE VELOCITY MATCHING

To further assess the effectiveness of StableVM, we adopt SiT-XL/2 (Ma et al., 2024) as the backbone model and train the model on ImageNet (resized to $256 \times 256$) (Deng et al., 2009). We train the models for 1M iterations with a batch size of 256. The baseline is the standard CFM, while our approach replaces the training objective with StableVM under different per-class bank capacities $K$. Following prior work (Ma et al., 2024; Yu et al., 2024), we employ the Euler-Maruyama SDE sampler with $w_t = \sigma_t$ and fix the number of function evaluations (NFEs) to 250 for all methods to ensure a fair comparison. We also apply classifier-free guidance with a scale of $w = 1.35$ and interval-based scheduling (Kynkäänniemi et al., 2024). For evaluation, we report FID (Heusel et al., 2017), IS (Salimans et al., 2016), sFID (Nash et al., 2021), precision, and recall (Kynkäänniemi et al., 2019).

From Table 1, we see that StableVM consistently outperforms the baseline CFM across different training stages, yielding a lower FID and higher IS while maintaining comparable precision and recall. The improvements are particularly evident especially at earlier and mid-training stages (250k–750k iterations), where larger bank capacities (*e.g.*, $K = 1024$) provide stronger gains. This indicates that a richer memory bank offers more stable and representative velocity targets, thereby accelerating and improving convergence.

**Two-Stage Training and Sampling.** To further validate the effectiveness of StableVM in *high-variance regime*, we propose to focus on only training $t$ in *high-variance regime*, and use a pretrained model for sampling during *low-variance regime*. This two-stage design is also used in Wan2.2 (Wan et al., 2025), highlighting its scalability and potential for performance gains. In this setting, our StableVM objective naturally serves as an ideal choice for variance reduction. Concretely, we sample $t \in [\xi, 1]$ for training and compare the CFM loss with our StableVM loss (bank capacity $K = 256$).

At inference, the model trained on the high-variance regime is used for $t \in [\xi, 1]$, while a pre-trained SiT-XL/2 model is used for $t \in [0, \xi)$. Fig. 3 reports FID versus training steps, showing consistent improvements with StableVM.

Additional details and results, including experimental setup, ablation studies, and unconditional generation results on synthetic GMMs, are provided in Appendix E.1.

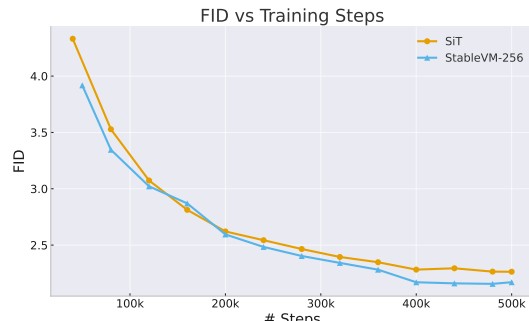

## 4.2 EVALUATION ON STABLE VELOCITY SAMPLING

Figure 3: Plot of FID for the two-stage experiments. Each model is trained up to 500k steps, and the checkpoints are evaluated every 40k steps. The data used for the plots can be found in Table 3.

We next evaluate the effectiveness of our proposed StableVS on large-scale pretrained text-to-image models. Fig. 4 presents qualitative comparisons using SD3.5 (Esser et al., 2024), where we show outputs from the default sampler with 50 and 25 steps, as well as our 25-step sampler under the same random seed. Compared to the default 25-step outputs, our method produces generations that remain visually closer to the 50-step references, preserving fine-grained details and semantic consistency even under a reduced step budget. Table 2 further quantifies this observation on *GenEval* (Ghosh et al., 2023) at $1024 \times 1024$ resolution across multiple pretrained models, including Flux-dev (Labs, 2024), SD3.5-Large and SD3-Medium (Esser et al., 2024). On non-reference metrics, our 25-step sampler consistently achieves performance closer to or surpassing the 50-step results compared to the default 25-step sampler. On reference metrics, StableVS provides dramatic improvements: PSNR nearly doubles and SSIM approaches the 50-step quality, confirming that our outputs remain perceptually closer to the ground-truth 50-step samples. Similar trends hold for Flux-dev and SD3-Medium, and these results validate that we can cut off redundant sampling steps in the *low-variance regime*. Additional experimental details and results are provided in Appendix E.2.

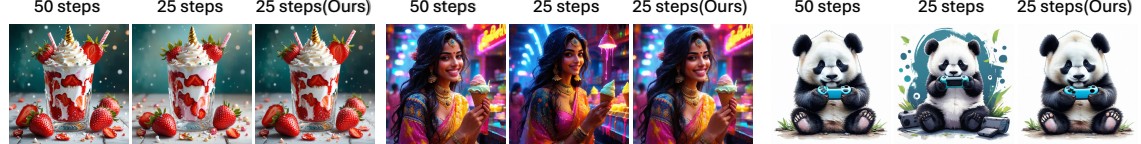

Figure 4: Visual comparisons of different prompts with SD3.5 (Esser et al., 2024). We show results using the default sampler with 50 and 25 steps, alongside our 25-step sampler under the same random seed. Compared to the default 25-step sampler, our method more closely aligns with the 50-step outputs. Zoom in for finer details. Additional qualitative comparisons are provided in Appendix G.

## 5 RELATED WORKS

**Training Acceleration of Diffusion/Flow Models.** Diffusion and flow matching models exhibit remarkable generative capabilities but incur substantial computational costs during training on high-resolution data. To mitigate this burden, prior work has pursued three complementary strategies: First, dimensionality reduction techniques aim to streamline training by compressing high-resolution data into lower-dimensional representations. Rombach et al. (2022) pioneered latent-space diffusion training, later enhanced by Chen et al. (2025) through improved compression ratios. Parallel efforts, such as the patch-wise score matching framework by Wang et al. (2023), further localized computational savings. A second direction integrates regularization with self-supervised learning objectives, as explored in Zheng et al. (2023); Yu et al. (2024), where auxiliary tasks stabilize diffusion model optimization. The third strategy focuses on improving the loss formulation. Studies like Karras et al. (2022b); Choi et al. (2022) address temporal inconsistencies in

Table 2: Evaluation results of multiple pretrained models on *GenEval* at $1024 \times 1024$ resolution, comparing the original method and ours. Metrics include non-reference metrics (overall, single-object, two-object, counting, colors, position, and color attribution) and reference metrics (PSNR, SSIM, LPIPS). Arrows ($\uparrow$ / $\downarrow$) indicate whether higher or lower values are better.

| Model | Non-reference metrics | | | | | | | Reference metrics | | |
|---|---|---|---|---|---|---|---|---|---|---|
| | Overall $\uparrow$ | Single $\uparrow$ | Two $\uparrow$ | Counting $\uparrow$ | Colors $\uparrow$ | Position $\uparrow$ | Color Attr $\uparrow$ | PSNR $\uparrow$ | SSIM $\uparrow$ | LPIPS $\downarrow$ |
| *SD3.5-Large* (Esser et al., 2024) | | | | | | | | | | |
| 50steps | **0.724** | **0.997** | 0.904 | **0.734** | 0.819 | 0.298 | 0.593 | – | – | – |
| 25steps | 0.711 | **0.997** | 0.886 | 0.675 | **0.832** | 0.295 | 0.583 | 15.576 | 0.735 | 0.370 |
| 25steps (Ours) | 0.718 | 0.994 | **0.904** | 0.706 | 0.806 | **0.300** | **0.600** | **30.954** | **0.958** | **0.050** |
| *Flux-dev* (Labs, 2024) | | | | | | | | | | |
| 50steps | 0.662 | 0.991 | 0.796 | **0.719** | 0.782 | 0.218 | 0.468 | – | – | – |
| 25steps | **0.663** | 0.991 | 0.816 | 0.703 | **0.782** | 0.208 | **0.478** | 18.406 | 0.794 | 0.281 |
| 25steps (Ours) | 0.657 | 0.988 | **0.816** | 0.706 | 0.766 | **0.218** | 0.450 | **29.187** | **0.937** | **0.068** |
| *SD3-medium* (Esser et al., 2024) | | | | | | | | | | |
| 50steps | **0.719** | 1.000 | **0.881** | **0.631** | 0.859 | **0.328** | 0.618 | – | – | – |
| 25steps | 0.703 | 0.994 | 0.864 | 0.578 | **0.870** | 0.318 | 0.593 | 14.872 | 0.416 | 0.680 |
| 25steps (Ours) | 0.713 | 0.997 | 0.861 | 0.625 | 0.854 | 0.315 | **0.625** | **28.493** | **0.940** | **0.077** |

training targets by proposing log-normal timestep sampling and signal-to-noise ratio (SNR)-aware weighting. Subsequent work Xu et al. (2023); Niedoba et al. (2024) employs self-normalized importance sampling (SNIS) estimators (Hesterberg, 1995) to reduce the variance of score estimation at the expense of bias. In this work, we extend the latter paradigm to the stochastic interpolants framework, introduce conditional mixture probabilities to remove bias, and demonstrate its efficacy in class-conditioned generation.

**Sampling Acceleration of Diffusion/Flow Models.** Reducing the number of sampling steps has become a key focus for both practical efficiency and theoretical analysis. Existing acceleration strategies can be broadly categorized into *training-required* and *training-free* approaches. Training-required methods typically aim to leverage additional learning to reduce steps. One approach is to distill pretrained multi-step diffusion models into few-step models (Salimans & Ho, 2022; Sauer et al., 2024). Other methods, such as consistency models (Song et al., 2023) and inductive moment matching (Zhou et al., 2025), enforce self-consistency of network outputs across different time steps, obviating the need for explicit distillation. Mean flows (Geng et al., 2025) take a related approach by predicting the average velocity along a trajectory, enabling one-step sampling. Training-free approaches, on the other hand, focus on designing more accurate numerical solvers for the reverse ODE. DDIM (Song et al., 2021) formulates sampling as a deterministic ODE and applies first-order integration, while high-order solvers such as DPM-Solver (Lu et al., 2022) and DPM-Solver++ (Lu et al., 2025) exploit the semi-linear structure to achieve higher accuracy in few-step generation. UniPC (Zhao et al., 2023) further unifies predictor–corrector schemes for diverse noise schedules, significantly reducing the quality gap relative to full-trajectory sampling. Our stable velocity sampling method is training-free, using analytical simplification in *low-variance regime* to reduce steps and remains compatible with other training-free solvers by employing them in *high-variance regime*.

## 6 CONCLUSION

In this work, we present a variance-based perspective on stochastic interpolants and uncover a two-regime structure that impacts both training and sampling. In the *high-variance regime*, conditional velocity estimates exhibit large variances, causing unstable and inefficient optimization. In contrast, in the *low-variance regime*, the estimator closely aligns with the true velocity, enabling analytical simplifications that accelerate sampling. Building on these insights, we introduce the **Stable Velocity** framework, which comprises: i) **Stable Velocity Matching (StableVM)**, a variance-reduced training objective that stabilizes optimization in the high-variance regime; ii) **Stable Velocity Sampling (StableVS)**, which leverages the low-variance regime to achieve faster, high-fidelity generation. Extensive evaluations on standard benchmarks and diverse pretrained models show that our approach improves both training convergence and sampling efficiency.

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

## A CONNECTIONS TO VARIANCE-PRESERVING DIFFUSION MODEL

In diffusion models (Sohl-Dickstein et al., 2015; Ho et al., 2020; Song et al., 2020), the forward process can be modeled as an Itô SDE:

$$d\boldsymbol{x} = \boldsymbol{f}(\boldsymbol{x}, t)\, dt + g(t)\, d\mathbf{w}, \tag{16}$$

where $\mathbf{w}$ is the standard Wiener process, $\boldsymbol{f}(\cdot, t) : \mathbb{R}^d \to \mathbb{R}^d$ is vector-valued function called drift coefficient of $x_t$ and $g(\cdot) : \mathbb{R} \to \mathbb{R}$ is a scalar function known as the diffusion coefficient of $\boldsymbol{x}_t$. It gradually transforms the data distribution $q$ to a known prior as time goes from $t = 0$ to 1. With stochastic process defined in Eq. (16), Variance-Preserving (VP) diffusion model (Ho et al., 2020; Song et al., 2020; Karras et al., 2022a) implicitly define both $\alpha_t$ and $\sigma_t$ in Eq. (1) with an equilibrium distribution as prior. VP diffusion commonly chooses $\alpha_t = \cos(\frac{\pi}{2}t), \sigma_t = \sin(\frac{\pi}{2}t)$.

The diffusion models is trained to estimate the score of the marginal at time t, $\nabla_{\boldsymbol{x}_t} p_t(\boldsymbol{x}_t)$, via a neural network. Specifically, the training objective is a weighted sum of the denoising score matching (DSM) (Vincent, 2011):

$$\min_{\boldsymbol{\theta}} \mathbb{E}_{t, q(\boldsymbol{x}_0), p_t(\boldsymbol{x}_t|\boldsymbol{x}_0)} \lambda_t \left\| \boldsymbol{s}_{\boldsymbol{\theta}}(\boldsymbol{x}_t, t) - \boldsymbol{s}_t(\boldsymbol{x}_t \mid \boldsymbol{x}_0) \right\|^2, \tag{17}$$

where $\lambda_t$ is a positive weighting function, $\boldsymbol{s}_{\boldsymbol{\theta}}(\cdot, \cdot) : [0, 1] \times \mathbb{R}^d \to \mathbb{R}^d$ is a time-dependent vector field parametrized as neural network with parameters $\boldsymbol{\theta}$, the conditional probability path $p_t(\boldsymbol{x}_t \mid \boldsymbol{x}_0) = \mathcal{N}(\boldsymbol{x}_t \mid \alpha_t \boldsymbol{x}_0, \sigma_t^2 \boldsymbol{I})$, and the conditional score function $\boldsymbol{s}_t(\boldsymbol{x}_t \mid \boldsymbol{x}_0) = \nabla_{\boldsymbol{x}_t} \log p_t(\boldsymbol{x}_t \mid \boldsymbol{x}_0)$. The Eq. (17) shares a very similar form with CFM target in Eq. (7). Also, similar to Eq. (8), the objective in Eq. (17) admits a closed-form minimizer (Song et al., 2020; Xu et al., 2023):

$$\boldsymbol{s}_{\boldsymbol{\theta}}^*(\boldsymbol{x}_t, t) = \mathbb{E}_{p_t(\boldsymbol{x}_0|\boldsymbol{x}_t)} \left[ \boldsymbol{s}_t(\boldsymbol{x}_t \mid \boldsymbol{x}_0) \right] = \boldsymbol{s}_t(\boldsymbol{x}_t) \tag{18}$$

Here, the marginal probability path $p_t(\boldsymbol{x}_t)$ is a mixture of conditional probability paths $p_t(\boldsymbol{x}_t \mid \boldsymbol{x}_0)$ that vary with data points $\boldsymbol{x}_0$, that is,

$$p_t(\boldsymbol{x}_t) = \int p_t(\boldsymbol{x}_t \mid \boldsymbol{x}_0) q(\boldsymbol{x}_0)\, d\boldsymbol{x}_0. \tag{19}$$

## B COMPARISON WITH STABLE TARGET FIELD

The standard training objective for diffusion models (Ho et al., 2020; Song et al., 2020; Karras et al., 2022b) is based on *denoising score matching* (DSM) (Vincent, 2011), also suffers from high variance.

To address this issue, Xu et al. (2023) proposed the *Stable Target Field* (STF), which stabilizes training by leveraging a reference batch $\mathcal{B} = \{\boldsymbol{x}_0^i\}_{i=1}^n \sim q(\boldsymbol{x}_0)$. The STF objective is defined as

$$\mathcal{L}_{\text{STF}}(\boldsymbol{\theta}, t) = \mathbb{E}_{\{\boldsymbol{x}_0^i\}_{i=1}^n \sim q(\boldsymbol{x}_0), p_t(\boldsymbol{x}_t|\boldsymbol{x}_0^1)} \left\| \boldsymbol{v}_{\boldsymbol{\theta}}(\boldsymbol{x}_t, t) - \sum_{k=1}^n \frac{p_t(\boldsymbol{x}_t \mid \boldsymbol{x}_0^k)}{\sum_{j=1}^n p_t(\boldsymbol{x}_t \mid \boldsymbol{x}_0^j)} \boldsymbol{s}_t(\boldsymbol{x}_t \mid \boldsymbol{x}_0^k) \right\|^2. \tag{20}$$

Unlike DSM ($n = 1$), STF forms a weighted average of scores over the reference batch, with weights determined by the conditional likelihoods $p_t(\boldsymbol{x}_t \mid \boldsymbol{x}_0^k)$. This reduces the covariance of the target by a factor of $n$, thereby lowering variance. While STF introduces bias, the minimizer of $\mathcal{L}_{\text{STF}}$ is given by

$$\boldsymbol{v}^*(\boldsymbol{x}_t, t) = \mathbb{E}_{p(\boldsymbol{x}_0|\boldsymbol{x}_t), \{\boldsymbol{x}_0^i\}_{i=2}^n \sim q(\boldsymbol{x}_0)} \sum_{k=1}^n \frac{p_t(\boldsymbol{x}_t \mid \boldsymbol{x}_0^k)}{\sum_{j=1}^n p_t(\boldsymbol{x}_t \mid \boldsymbol{x}_0^j)} \boldsymbol{s}_t(\boldsymbol{x}_t \mid \boldsymbol{x}_0^k), \tag{21}$$

which deviates from the true score $\boldsymbol{s}_t(\boldsymbol{x}_t)$. However, as $n \to \infty$, this bias vanishes and the weighted estimator converges to the true score.

Our approach differs from STF in three important ways:

1. **General framework.** We extend the variance analysis and variance-reduction strategy to the flow matching as well as stochastic interpolant framework, which generalizes beyond VP diffusion and exhibits a distinct variance structure.

2. **Unbiased objective.** Instead of relying on a finite-sample weighted average, we propose a mixture of conditional probabilities that eliminates bias while still achieving variance reduction.

3. **Class-conditional extension.** While STF does not naturally extend to class-conditional settings, we design a tailored algorithm that maintains variance reduction under classifier-free guidance, improving both convergence and training efficiency.

---

**Algorithm 2** Stable Velocity Matching with Classifier-free Guidance

---

**Require:** Training iterations $T$, model $\boldsymbol{v_\theta}$, dataset $\mathcal{D}$, learning rate $\eta$, batch size $B$, number of classes $C$, per-class bank capacity $K$, CFG dropout prob $p_{\mathrm{cfg}}$

**Initialize memory bank** $\mathcal{M} = \{\mathcal{M}_c\}_{c=0}^C$ $\qquad \triangleright c=0,\ldots,C-1$ are classes; $c=C$ is the *unconditional* bucket

**for** $c = 0,\ldots,C$ **do**
$\quad \mathcal{M}_c \leftarrow$ Prefilled FIFO queue of capacity $K$
**end for**

**for** $iter = 1 \ldots T$ **do**
$\quad$ Sample times $\{t_i\}_{i=1}^B \sim q_t([0,1])$
$\quad$ Uniformly sample input labels $\{y^i\}_{i=1}^B$ from $0,\ldots,C-1$
$\quad y^i \leftarrow C$ with probability $p_{\mathrm{cfg}}$

$\rule{6cm}{0.4pt}$ **Stable Velocity Matching Update** $\rule{6cm}{0.4pt}$

$\quad$ **for** $i = 1 \ldots B$ **do**
$\qquad$ Sample perturbed samples $\boldsymbol{x}_{t_i}^i$ from

$$p_{\mathrm{GMM}}(\boldsymbol{x}_{t_i}^i \mid \mathcal{M}_{y^i}) = \frac{1}{|\mathcal{M}_{y^i}|} \sum_{\boldsymbol{x}_0^{\mathrm{ref}} \in \mathcal{M}_{y^i}} p_t(\boldsymbol{x}_{t_i}^j \mid \boldsymbol{x}_0^{\mathrm{ref}})$$

$\qquad$ Compute stable field

$$\boldsymbol{v}_{\mathcal{M}_{y^i}}(\boldsymbol{x}_{t_i}^j) = \sum_{\boldsymbol{x}_0^{\mathrm{ref}} \in \mathcal{M}_{y^i}} \frac{p_t(\boldsymbol{x}_{t_i}^j \mid \boldsymbol{x}_0^{\mathrm{ref}})}{\sum_{\boldsymbol{y}_0^{\mathrm{ref}} \in \mathcal{M}_{y^i}} p_t(\boldsymbol{x}_{t_i}^j \mid \boldsymbol{y}_0^{\mathrm{ref}})} \, \boldsymbol{v}_t(\boldsymbol{x}_{t_i}^j \mid \boldsymbol{x}_0^{\mathrm{ref}})$$

$\quad$ **end for**

$\quad$ Calculate loss for $\boldsymbol{v_\theta}$:

$$\mathcal{L}(\boldsymbol{\theta}) = \frac{1}{B} \sum_{j=1}^B \lambda(t_i) \left\| \boldsymbol{v_\theta}(\boldsymbol{x}_{t_i}^j, t_i, y^j) - \boldsymbol{v}_{\mathcal{M}_{y^i}}(\boldsymbol{x}_{t_i}^j) \right\|^2$$

$\quad \boldsymbol{\theta} \leftarrow \boldsymbol{\theta} - \eta \nabla_{\boldsymbol{\theta}} \mathcal{L}(\boldsymbol{\theta})$

$\rule{6cm}{0.4pt}$ **Online Memory Bank Update** $\rule{6cm}{0.4pt}$

$\quad$ Sample $\mathcal{B} = \{(\boldsymbol{x}_0^i, y_i)\}_{i=1}^B$ from $\mathcal{D}$
$\quad$ **for** $i = 1,\ldots,B$ **do**
$\qquad$ Push $\boldsymbol{x}_0^i$ to $\mathcal{M}_{y_i}$ (evict oldest if full)
$\qquad$ Push $\boldsymbol{x}_0^i$ to $\mathcal{M}_C$ $\qquad\qquad\qquad\qquad\qquad \triangleright$ Unconditional bucket stores all samples
$\quad$ **end for**
**end for**
**return** $\boldsymbol{v_\theta}$

---

## C  EXTENSION TO CLASS-CONDITIONAL GENERATION

We further extend StableVM to the class-conditional generation setting, with the complete procedure summarized in Algorithm 2.

## D  PROOFS

### D.1  $p_t^{\text{GMM}}$ POSTERIOR (PROPOSITION 1)

**Proposition 1.** *The posterior $p_t^{GMM}\left(\{x_0^i\}_{i=1}^n \mid x_t\right) = \frac{1}{n}\sum_{i=1}^n \left(p_t\left(x_0^i \mid x_t\right)\prod_{j\neq i} q(x_0^j)\right)$.*

*Proof.* This is a simple application of the Bayes rule. Note that we have

$$
\begin{aligned}
p_t^{\text{GMM}}\left(\{x_0^i\}_{i=1}^n \mid x_t\right) &= \frac{p_t^{\text{GMM}}\left(x_t \mid \{x_0^i\}_{i=1}^n\right) \cdot \prod_{i=1}^n q(x_0^i)}{p_t(x_t)} \\
&= \frac{1}{p_t(x_t)}\left(\sum_{i=1}^n \frac{1}{n}\cdot p_t(x_t \mid x_0^i)\right)\left(\prod_{i=1}^n q(x_0^i)\right) \\
&= \frac{1}{n}\sum_{i=1}^n \left(\frac{p_t(x_t \mid x_0^i)}{p_t(x_t)}\cdot \prod_{j=1}^n q(x_0^j)\right) \\
&= \frac{1}{n}\sum_{i=1}^n \left(p_t(x_0^i \mid x_t)\prod_{j\neq i} q(x_0^j)\right),
\end{aligned}
\tag{22}
$$

as desired.  □

### D.2  PROOF OF UNBIASEDNESS (THEOREM 1)

**Theorem 1.** *(a) The StableVM target is unbiased. That is, for any $x_t$, we have*

$$
\mathbb{E}_{\{x_0^i\}_{i=1}^n \sim p_t^{GMM}(\cdot \mid x_t)}\left[\sum_{k=1}^n \frac{p_t(x_t \mid x_0^k)v_t(x_t \mid x_0^k)}{\sum_{j=1}^n p_t(x_t \mid x_0^j)}\right] = v_t(x_t).
$$

*(b) The global minimizer $v^*(x_t, t)$ of the StableVM objective $\mathcal{L}_{StableVM}$ is the true velocity field $v_t(x_t)$.*

*Proof.* (a) Using Eq. (22), we obtain

$$
\mathbb{E}_{\{\boldsymbol{x}_0^i\}_{i=1}^n \sim p_t^{\text{GMM}}(\cdot|\boldsymbol{x}_t)} \left[ \sum_{k=1}^n \frac{p_t(\boldsymbol{x}_t \mid \boldsymbol{x}_0^k)\boldsymbol{v}_t(\boldsymbol{x}_t \mid \boldsymbol{x}_0^k)}{\sum_{j=1}^n p_t(\boldsymbol{x}_t \mid \boldsymbol{x}_0^j)} \right]
$$

$$
= \int \frac{1}{n \cdot p_t(\boldsymbol{x}_t)} \left( \sum_{i=1}^n p_t(\boldsymbol{x}_t \mid \boldsymbol{x}_0^i) \right) \left( \prod_{i=1}^n q(\boldsymbol{x}_0^i) \right) \cdot \sum_{k=1}^n \frac{p_t(\boldsymbol{x}_t \mid \boldsymbol{x}_0^k)\boldsymbol{v}_t(\boldsymbol{x}_t \mid \boldsymbol{x}_0^k)}{\sum_{j=1}^n p_t(\boldsymbol{x}_t \mid \boldsymbol{x}_0^j)} \, \mathrm{d}\boldsymbol{x}_0^{1:n}
$$

$$
= \frac{1}{n \cdot p_t(\boldsymbol{x}_t)} \int \left( \prod_{i=1}^n q(\boldsymbol{x}_0^i) \right) \left( \sum_{k=1}^n p_t(\boldsymbol{x}_t \mid \boldsymbol{x}_0^k)\boldsymbol{v}_t(\boldsymbol{x}_t \mid \boldsymbol{x}_0^k) \right) \, \mathrm{d}\boldsymbol{x}_0^{1:n}
$$

$$
= \frac{1}{n \cdot p_t(\boldsymbol{x}_t)} \sum_{k=1}^n \int \left( \prod_{i=1}^n q(\boldsymbol{x}_0^i) \right) p_t(\boldsymbol{x}_t \mid \boldsymbol{x}_0^k)\boldsymbol{v}_t(\boldsymbol{x}_t \mid \boldsymbol{x}_0^k) \, \mathrm{d}\boldsymbol{x}_0^{1:n}
$$

$$
= \frac{1}{n \cdot p_t(\boldsymbol{x}_t)} \sum_{k=1}^n \int \left( \prod_{i \neq k} q(\boldsymbol{x}_0^i) \right) p_t(\boldsymbol{x}_t, \boldsymbol{x}_0^k)\boldsymbol{v}_t(\boldsymbol{x}_t \mid \boldsymbol{x}_0^k) \, \mathrm{d}\boldsymbol{x}_0^{1:n}
$$

$$
= \frac{1}{n \cdot p_t(\boldsymbol{x}_t)} \sum_{k=1}^n \left( \left( \prod_{i \neq k} \int q(\boldsymbol{x}_0^i) \, \mathrm{d}\boldsymbol{x}_0^i \right) \cdot \int p_t(\boldsymbol{x}_t, \boldsymbol{x}_0^k)\boldsymbol{v}_t(\boldsymbol{x}_t \mid \boldsymbol{x}_0^k) \, \mathrm{d}\boldsymbol{x}_0^k \right)
$$

$$
= \frac{1}{n} \sum_{k=1}^n \int p_t(\boldsymbol{x}_0^k \mid \boldsymbol{x}_t)\boldsymbol{v}_t(\boldsymbol{x}_t \mid \boldsymbol{x}_0^k) \, \mathrm{d}\boldsymbol{x}_0^k = \boldsymbol{v}_t(\boldsymbol{x}_t),
$$

as desired.

(b) Recall the StableVM objective Eq. (10)

$$
\mathcal{L}_{\text{StableVM}}(\boldsymbol{\theta}, t)
$$

$$
= \mathbb{E}_{\boldsymbol{x}_t \sim p_t, \, \{\boldsymbol{x}_0^i\}_{i=1}^n \sim p_t^{\text{GMM}}(\cdot|\boldsymbol{x}_t)} \left[ \left\| \boldsymbol{v}_{\boldsymbol{\theta}}(\boldsymbol{x}_t, t) - \sum_{k=1}^n \frac{p_t(\boldsymbol{x}_t \mid \boldsymbol{x}_0^k)\boldsymbol{v}_t(\boldsymbol{x}_t \mid \boldsymbol{x}_0^k)}{\sum_{j=1}^n p_t(\boldsymbol{x}_t \mid \boldsymbol{x}_0^j)} \right\|^2 \right]
$$

$$
= \mathbb{E}_{\boldsymbol{x}_t \sim p_t} \left[ \int p_t^{\text{GMM}} \left( \{\boldsymbol{x}_0^i\}_{i=1}^n \mid \boldsymbol{x}_t \right) \left\| \boldsymbol{v}_{\boldsymbol{\theta}}(\boldsymbol{x}_t, t) - \sum_{k=1}^n \frac{p_t(\boldsymbol{x}_t \mid \boldsymbol{x}_0^k)\boldsymbol{v}_t(\boldsymbol{x}_t \mid \boldsymbol{x}_0^k)}{\sum_{j=1}^n p_t(\boldsymbol{x}_t \mid \boldsymbol{x}_0^j)} \right\|^2 \, \mathrm{d}\boldsymbol{x}_0^{1:n} \right].
$$

For each $\boldsymbol{x}_t$, let

$$
\mathcal{L}_{\boldsymbol{x}_t}(\boldsymbol{v}) := \int p_t^{\text{GMM}} \left( \{\boldsymbol{x}_0^i\}_{i=1}^n \mid \boldsymbol{x}_t \right) \left\| \boldsymbol{v} - \sum_{k=1}^n \frac{p_t(\boldsymbol{x}_t \mid \boldsymbol{x}_0^k)\boldsymbol{v}_t(\boldsymbol{x}_t \mid \boldsymbol{x}_0^k)}{\sum_{j=1}^n p_t(\boldsymbol{x}_t \mid \boldsymbol{x}_0^j)} \right\|^2 \, \mathrm{d}\boldsymbol{x}_0^{1:n}.
$$

It suffices to show that for each $\boldsymbol{x}_t$, setting $\boldsymbol{v}$ to $\boldsymbol{v}_t(\boldsymbol{x}_t)$ above minimizes $\mathcal{L}_{\boldsymbol{x}_t}$. Note that $\mathcal{L}_{\boldsymbol{x}_t}$ is differentiable and strictly convex in $\boldsymbol{v}$, so the minimizer $\boldsymbol{v}^*$ must satisfy $\nabla \mathcal{L}_{\boldsymbol{x}_t}(\boldsymbol{v}^*) = 0$. It follows that

$$
\begin{aligned}
0 &= 2 \int p_t^{\mathrm{GMM}}\left(\{\boldsymbol{x}_0^i\}_{i=1}^n \mid \boldsymbol{x}_t\right) \left(\boldsymbol{v}^* - \sum_{k=1}^n \frac{p_t(\boldsymbol{x}_t \mid \boldsymbol{x}_0^k) \boldsymbol{v}_t(\boldsymbol{x}_t \mid \boldsymbol{x}_0^k)}{\sum_{j=1}^n p_t(\boldsymbol{x}_t \mid \boldsymbol{x}_0^j)}\right) \, \mathrm{d}\boldsymbol{x}_0^{1:n} \\
&= 2 \int p_t^{\mathrm{GMM}}\left(\{\boldsymbol{x}_0^i\}_{i=1}^n \mid \boldsymbol{x}_t\right) \boldsymbol{v}^* \, \mathrm{d}\boldsymbol{x}_0^{1:n} \\
&\quad - 2 \int p_t^{\mathrm{GMM}}\left(\{\boldsymbol{x}_0^i\}_{i=1}^n \mid \boldsymbol{x}_t\right) \sum_{k=1}^n \frac{p_t(\boldsymbol{x}_t \mid \boldsymbol{x}_0^k) \boldsymbol{v}_t(\boldsymbol{x}_t \mid \boldsymbol{x}_0^k)}{\sum_{j=1}^n p_t(\boldsymbol{x}_t \mid \boldsymbol{x}_0^j)} \, \mathrm{d}\boldsymbol{x}_0^{1:n} \\
&= 2\boldsymbol{v}^* - 2\boldsymbol{v}_t(\boldsymbol{x}_t),
\end{aligned}
$$

where we have used part (a) in the last step. Therefore, $\boldsymbol{v}_t(\boldsymbol{x}_t)$ is the unique minimizer of $\mathcal{L}_{\boldsymbol{x}_t}$. This finishes the proof.

$\square$

### D.3 PROOFS OF THE VARIANCE BOUNDS (THEOREM 2 AND THEOREM 3)

In this section, we prove the variance reduction bound of our StableVM target as in Eq. (11). We first recall that the StableVM target is the following estimator

$$
\widehat{\boldsymbol{v}}_t := \sum_{k=1}^n \frac{p_t(\boldsymbol{x}_t \mid \boldsymbol{x}_0^k) \, \boldsymbol{v}_t(\boldsymbol{x}_t \mid \boldsymbol{x}_0^k)}{\sum_{j=1}^n p_t(\boldsymbol{x}_t \mid \boldsymbol{x}_0^j)} \in \mathbb{R}^d, \tag{23}
$$

where $\boldsymbol{x}_t \sim p_t$ and $\{\boldsymbol{x}_0^i\}_{i=1}^n \sim p_t^{\mathrm{GMM}}(\cdot \mid \boldsymbol{x}_t)$.

We first prove the weaker version of the variance bound.

**Theorem 2.** *Fix $t \in [0, 1]$. We always have $\mathcal{V}_{StableVM}(t) \leq \mathcal{V}_{CFM}(t)$.*

*Proof.* We have

$$\mathcal{V}_{\text{StableVM}}(t)$$

$$= \mathbb{E}_{\{\boldsymbol{x}_0^i\}_{i=1}^n \sim q^n} \mathbb{E}_{\boldsymbol{x}_t \sim p_t^{\text{GMM}}(\cdot | \{\boldsymbol{x}_0^i\}_{i=1}^n)} \left\| \boldsymbol{v}_t(\boldsymbol{x}_t) - \sum_{k=1}^n \frac{p_t(\boldsymbol{x}_t \mid \boldsymbol{x}_0^k) \boldsymbol{v}_t(\boldsymbol{x}_t \mid \boldsymbol{x}_0^k)}{\sum_{j=1}^n p_t(\boldsymbol{x}_t \mid \boldsymbol{x}_0^j)} \right\|^2$$

$$= \mathbb{E}_{\{\boldsymbol{x}_0^i\}_{i=1}^n \sim q^n} \left[ \int \frac{1}{n} \left( \sum_{i=1}^n p_t(\boldsymbol{x}_t \mid \boldsymbol{x}_0^i) \right) \left\| \boldsymbol{v}_t(\boldsymbol{x}_t) - \sum_{k=1}^n \frac{p_t(\boldsymbol{x}_t \mid \boldsymbol{x}_0^k) \boldsymbol{v}_t(\boldsymbol{x}_t \mid \boldsymbol{x}_0^k)}{\sum_{j=1}^n p_t(\boldsymbol{x}_t \mid \boldsymbol{x}_0^j)} \right\|^2 \mathrm{d}\boldsymbol{x}_t \right]$$

$$\leq \mathbb{E}_{\{\boldsymbol{x}_0^i\}_{i=1}^n \sim q^n} \left[ \int \frac{1}{n} \left( \sum_{i=1}^n p_t(\boldsymbol{x}_t \mid \boldsymbol{x}_0^i) \right) \sum_{k=1}^n \frac{p_t(\boldsymbol{x}_t \mid \boldsymbol{x}_0^k)}{\sum_{j=1}^n p_t(\boldsymbol{x}_t \mid \boldsymbol{x}_0^j)} \left\| \boldsymbol{v}_t(\boldsymbol{x}_t) - \boldsymbol{v}_t(\boldsymbol{x}_t \mid \boldsymbol{x}_0^k) \right\|^2 \mathrm{d}\boldsymbol{x}_t \right]$$

$$= \mathbb{E}_{\{\boldsymbol{x}_0^i\}_{i=1}^n \sim q^n} \left[ \int \frac{1}{n} \sum_{k=1}^n p_t(\boldsymbol{x}_t \mid \boldsymbol{x}_0^k) \left\| \boldsymbol{v}_t(\boldsymbol{x}_t) - \boldsymbol{v}_t(\boldsymbol{x}_t \mid \boldsymbol{x}_0^k) \right\|^2 \mathrm{d}\boldsymbol{x}_t \right]$$

$$= \frac{1}{n} \cdot \mathbb{E}_{\{\boldsymbol{x}_0^i\}_{i=1}^n \sim q^n} \left[ \sum_{k=1}^n \int p_t(\boldsymbol{x}_t \mid \boldsymbol{x}_0^k) \left\| \boldsymbol{v}_t(\boldsymbol{x}_t) - \boldsymbol{v}_t(\boldsymbol{x}_t \mid \boldsymbol{x}_0^k) \right\|^2 \mathrm{d}\boldsymbol{x}_t \right]$$

$$= \frac{1}{n} \cdot \mathbb{E}_{\{\boldsymbol{x}_0^i\}_{i=1}^n \sim q^n} \left[ \sum_{k=1}^n \mathbb{E}_{\boldsymbol{x}_t \sim p_t(\cdot | \boldsymbol{x}_0^k)} \left\| \boldsymbol{v}_t(\boldsymbol{x}_t) - \boldsymbol{v}_t(\boldsymbol{x}_t \mid \boldsymbol{x}_0^k) \right\|^2 \right]$$

$$= \frac{1}{n} \sum_{k=1}^n \mathcal{V}_{\text{CFM}}(t) = \mathcal{V}_{\text{CFM}}(t),$$

where the inequality is by Jensen's inequality and the convexity of $\| \cdot \|^2$. $\qquad\square$

Before proceeding to proving the stronger bound, we first give an alternative interpretation of the posterior $p_t^{\text{GMM}} \left( \{\boldsymbol{x}_0^i\}_{i=1}^n \mid \boldsymbol{x}_t \right)$. Consider the following procedure conditioned on $\boldsymbol{x}_t$:

1. Sample a latent index $I$ uniformly from $\{1, \ldots, n\}$.

2. Sample $\boldsymbol{x}_0^I \sim p_t(\cdot \mid \boldsymbol{x}_t)$ and $\boldsymbol{x}_0^j \sim q$ for all $j \neq I$.

We claim the following:

**Lemma 1.** *(a) The joint distribution of $\{\boldsymbol{x}_0^i\}_{i=1}^n$ sampled from the above procedure conditioned on $\boldsymbol{x}_t$ is exactly $p_t^{GMM} \left( \{\boldsymbol{x}_0^i\}_{i=1}^n \mid \boldsymbol{x}_t \right)$.*

*(b) $\{\boldsymbol{x}_0^i\}_{i=1}^n$ are independent conditioned on $\boldsymbol{x}_t$ and $I$.*

*Proof.* (a) Note that the joint distribution of $\{\boldsymbol{x}_0^i\}_{i=1}^n$ sampled from the above procedure is

$$\frac{1}{n} \sum_{i=1}^n \left( p_t(\boldsymbol{x}_0^i \mid \boldsymbol{x}_t) \prod_{j \neq i} q(\boldsymbol{x}_0^j) \right),$$

which matches the joint distribution given by the posterior of $p_t^{\text{GMM}}$.

(b) This is clear by construction.

$\qquad\square$

The following lemmas compute the variance of the StableVM estimator step-by-step.

**Lemma 2.** *We have*
$$\mathrm{Cov}\left(\widehat{\boldsymbol{v}}_t \mid \boldsymbol{x}_t\right) = \mathbb{E}\left[\mathrm{Cov}\left(\widehat{\boldsymbol{v}}_t \mid \boldsymbol{x}_t, I\right) \mid \boldsymbol{x}_t\right],$$
*where the expectation on the right-hand side is over the random variable $I$.*

*Proof.* By the law of total covariance, we have
$$\mathrm{Cov}\left(\widehat{\boldsymbol{v}}_t \mid \boldsymbol{x}_t\right) = \mathbb{E}_I\left[\mathrm{Cov}\left(\widehat{\boldsymbol{v}}_t \mid \boldsymbol{x}_t, I\right) \mid \boldsymbol{x}_t\right] + \mathrm{Cov}_I\left(\mathbb{E}\left[\widehat{\boldsymbol{v}}_t \mid \boldsymbol{x}_t, I\right] \mid \boldsymbol{x}_t\right).$$
We claim that $\mathbb{E}\left[\widehat{\boldsymbol{v}}_t \mid \boldsymbol{x}_t, I\right]$ does not depend on $I$, so the second term above would be 0.

Let $i \in [n]$. We have
$$\mathbb{E}\left[\widehat{\boldsymbol{v}}_t \mid \boldsymbol{x}_t, I = i\right] = \int p_t(\boldsymbol{x}_0^i \mid \boldsymbol{x}_t) \prod_{j \neq i} q(\boldsymbol{x}_0^j) \cdot \sum_{k=1}^n \frac{p_t(\boldsymbol{x}_t \mid \boldsymbol{x}_0^k)\,\boldsymbol{v}_t(\boldsymbol{x}_t \mid \boldsymbol{x}_0^k)}{\sum_{j=1}^n p_t(\boldsymbol{x}_t \mid \boldsymbol{x}_0^j)}\, \mathrm{d}\boldsymbol{x}_0^{1:n}.$$

Let $\pi : \mathbb{R}^{nd} \to \mathbb{R}^{nd}$ that swaps $\boldsymbol{x}_0^1$ and $\boldsymbol{x}_0^i$, i.e. $\pi(\boldsymbol{x}_0^1, \ldots, \boldsymbol{x}_0^i, \ldots, \boldsymbol{x}_0^n) = (\boldsymbol{x}_0^i, \ldots, \boldsymbol{x}_0^1, \ldots, \boldsymbol{x}_0^n)$. Note that we have $|\det D\pi| = 1$ since the Jacobian $D\pi$ is a permutation matrix. Then by the change of variable formula, we have
$$\mathbb{E}\left[\widehat{\boldsymbol{v}}_t \mid \boldsymbol{x}_t, I = i\right] = \int p_t(\boldsymbol{x}_0^1 \mid \boldsymbol{x}_t) \prod_{j \neq 1} q(\boldsymbol{x}_0^j) \cdot \sum_{k=1}^n \frac{p_t(\boldsymbol{x}_t \mid \boldsymbol{x}_0^k)\,\boldsymbol{v}_t(\boldsymbol{x}_t \mid \boldsymbol{x}_0^k)}{\sum_{j=1}^n p_t(\boldsymbol{x}_t \mid \boldsymbol{x}_0^j)}\, \mathrm{d}\boldsymbol{x}_0^{1:n}$$
$$= \mathbb{E}\left[\widehat{\boldsymbol{v}}_t \mid \boldsymbol{x}_t, I = 1\right].$$
This shows that $\mathrm{Cov}_I\left(\mathbb{E}\left[\widehat{\boldsymbol{v}}_t \mid \boldsymbol{x}_t, I\right] \mid \boldsymbol{x}_t\right) = 0$, which finishes the proof. $\qquad\square$

For the next two lemmas, we fix $\ell \in [d]$ and only consider the $\ell^{\text{th}}$ component $\widehat{\boldsymbol{v}}_t^{(\ell)}$ for simplicity. Define
$$V_{n-1} := \sum_{k \neq i} p_t(\boldsymbol{x}_t \mid \boldsymbol{x}_0^k)\boldsymbol{v}_t^{(\ell)}(\boldsymbol{x}_t \mid \boldsymbol{x}_0^k), \quad P_{n-1} := \sum_{k \neq i} p_t(\boldsymbol{x}_t \mid \boldsymbol{x}_0^k),$$
$$v_i := p_t(\boldsymbol{x}_t \mid \boldsymbol{x}_0^i)\boldsymbol{v}_t^{(\ell)}(\boldsymbol{x}_t \mid \boldsymbol{x}_0^i), \quad p_i := p_t(\boldsymbol{x}_t \mid \boldsymbol{x}_0^i).$$
Then we have
$$\widehat{\boldsymbol{v}}_t^{(\ell)} = \frac{V_{n-1} + v_i}{P_{n-1} + p_i}.$$
Note that conditioned on $\boldsymbol{x}_t$ and $I = i$, the random variables $\boldsymbol{x}_0^k$ for $k \neq i$ are all independent and follow the data distribution $q$ by Lemma 1.

Let us also first compute some expectations that will be useful later. We have for $k \neq i$,
$$\mathbb{E}\left[p_t(\boldsymbol{x}_t \mid \boldsymbol{x}_0^k) \mid \boldsymbol{x}_t, I = i\right] = \int p_t(\boldsymbol{x}_t \mid \boldsymbol{x}_0^k)q(\boldsymbol{x}_0^k)\,\mathrm{d}\boldsymbol{x}_0^k = \int p_t(\boldsymbol{x}_t, \boldsymbol{x}_0^k)\,\mathrm{d}\boldsymbol{x}_0^k = p_t(\boldsymbol{x}_t), \qquad (24)$$

$$\mathbb{E}\left[p_t(\boldsymbol{x}_t \mid \boldsymbol{x}_0^k)\boldsymbol{v}_t(\boldsymbol{x}_t \mid \boldsymbol{x}_0^k) \mid \boldsymbol{x}_t, I = i\right] = \int p_t(\boldsymbol{x}_t \mid \boldsymbol{x}_0^k)\boldsymbol{v}_t(\boldsymbol{x}_t \mid \boldsymbol{x}_0^k)q(\boldsymbol{x}_0^k)\,\mathrm{d}\boldsymbol{x}_0^k$$
$$= \int p_t(\boldsymbol{x}_t, \boldsymbol{x}_0^k)\boldsymbol{v}_t(\boldsymbol{x}_t \mid \boldsymbol{x}_0^k)\,\mathrm{d}\boldsymbol{x}_0^k$$
$$= p_t(\boldsymbol{x}_t)\int \frac{p_t(\boldsymbol{x}_t, \boldsymbol{x}_0^k)}{p_t(\boldsymbol{x}_t)}\,\boldsymbol{v}_t(\boldsymbol{x}_t \mid \boldsymbol{x}_0^k)\,\mathrm{d}\boldsymbol{x}_0^k$$
$$= p_t(\boldsymbol{x}_t)\int p_t(\boldsymbol{x}_0^k \mid \boldsymbol{x}_t)\boldsymbol{v}_t(\boldsymbol{x}_t \mid \boldsymbol{x}_0^k)\,\mathrm{d}\boldsymbol{x}_0^k$$
$$= p_t(\boldsymbol{x}_t)\boldsymbol{v}_t(\boldsymbol{x}_t), \qquad (25)$$

$$\mathbb{E}_{\boldsymbol{x}_0 \sim p_t(\cdot | \boldsymbol{x}_t)} \left[ \boldsymbol{v}_t(\boldsymbol{x}_t \mid \boldsymbol{x}_0) \mid \boldsymbol{x}_t \right] = \int p_t(\boldsymbol{x}_0 \mid \boldsymbol{x}_t) \, \boldsymbol{v}_t(\boldsymbol{x}_t \mid \boldsymbol{x}_0) \, \mathrm{d}\boldsymbol{x}_0 = \boldsymbol{v}_t(\boldsymbol{x}_t), \tag{26}$$

and

$$\mathbb{E}_{\boldsymbol{x}_0 \sim q} \left[ \frac{p_t(\boldsymbol{x}_0 \mid \boldsymbol{x}_t)}{q(\boldsymbol{x}_0)} \right] = \int p_t(\boldsymbol{x}_0 \mid \boldsymbol{x}_t) \, \mathrm{d}\boldsymbol{x}_0 = 1. \tag{27}$$

We then continue with the lemmas.

**Lemma 3.** *As $n \to \infty$, we have*

$$\sqrt{n-1} \left( \frac{V_{n-1}}{P_{n-1}} - \boldsymbol{v}_t^{(\ell)}(\boldsymbol{x}_t) \right)$$

$$\xrightarrow{d} \mathcal{N} \left( 0, \mathbb{E}_{\boldsymbol{x}_0 \sim p_t(\cdot | \boldsymbol{x}_t)} \left[ \frac{p_t(\boldsymbol{x}_t \mid \boldsymbol{x}_0)}{p_t(\boldsymbol{x}_t)} \left( \boldsymbol{v}_t^{(\ell)}(\boldsymbol{x}_t \mid \boldsymbol{x}_0) - \boldsymbol{v}_t^{(\ell)}(\boldsymbol{x}_t) \right)^2 \right] \right),$$

*where the random variables $V_{n-1}$ and $P_{n-1}$ are conditioned on $\boldsymbol{x}_t$ and $I = i$.*

*Proof.* Write

$$f(\boldsymbol{x}_0) := \boldsymbol{v}_t^{(\ell)}(\boldsymbol{x}_t \mid \boldsymbol{x}_0), \quad w(\boldsymbol{x}_0) := \frac{p_t(\boldsymbol{x}_0 \mid \boldsymbol{x}_t)}{q(\boldsymbol{x}_0)}.$$

Then we have

$$\frac{V_{n-1}}{P_{n-1}} = \frac{\sum_{k \neq i} p_t(\boldsymbol{x}_t \mid \boldsymbol{x}_0^k) \, \boldsymbol{v}_t^{(\ell)}(\boldsymbol{x}_t \mid \boldsymbol{x}_0^k)}{\sum_{k \neq i} p_t(\boldsymbol{x}_t \mid \boldsymbol{x}_0^k)} = \frac{\sum_{k \neq i} p_t(\boldsymbol{x}_t \mid \boldsymbol{x}_0^k) \, f(\boldsymbol{x}_0^k)}{\sum_{k \neq i} p_t(\boldsymbol{x}_t \mid \boldsymbol{x}_0^k)}$$

$$= \frac{\sum_{k \neq i} \frac{p_t(\boldsymbol{x}_t | \boldsymbol{x}_0^k) \, q(\boldsymbol{x}_0^k)}{p_t(\boldsymbol{x}_t) \, q(\boldsymbol{x}_0^k)} f(\boldsymbol{x}_0^k)}{\sum_{k \neq i} \frac{p_t(\boldsymbol{x}_t | \boldsymbol{x}_0^k) \, q(\boldsymbol{x}_0^k)}{p_t(\boldsymbol{x}_t) \, q(\boldsymbol{x}_0^k)}} = \frac{\sum_{k \neq i} \frac{p_t(\boldsymbol{x}_0^k | \boldsymbol{x}_t)}{q(\boldsymbol{x}_0^k)} f(\boldsymbol{x}_0^k)}{\sum_{k \neq i} \frac{p_t(\boldsymbol{x}_0^k | \boldsymbol{x}_t)}{q(\boldsymbol{x}_0^k)}}$$

$$= \frac{\sum_{k \neq i} w(\boldsymbol{x}_0^k) \, f(\boldsymbol{x}_0^k)}{\sum_{k \neq i} w(\boldsymbol{x}_0^k)}. \tag{28}$$

Recall from Lemma 1 that conditioned on $\boldsymbol{x}_t$ and $I = i$, the $\boldsymbol{x}_0^i$'s are all independent. Therefore, Eq. (28) is a self-normalized importance sampling estimator (Chapter 9 of Owen (2013)) with importance distribution $q(\boldsymbol{x}_0)$, nominal distribution $p_t(\boldsymbol{x}_0 \mid \boldsymbol{x}_t)$, and importance weight ratio $w(\boldsymbol{x}_0)$.

Note that by Eq. (26) and Eq. (27), we have

$$\mathbb{E}_{\boldsymbol{x}_0 \sim p_t(\cdot | \boldsymbol{x}_t)}[f(\boldsymbol{x}_0)] = \boldsymbol{v}_t^{(\ell)}(\boldsymbol{x}_t), \quad \mathbb{E}_{\boldsymbol{x}_0 \sim q}[w(\boldsymbol{x}_0)] = 1.$$

Following results using the delta method as in Lehmann & Romano (2023) (Theorem 11.2.14) and Owen (2013) (Eq. (9.8)), we have that

$$\sqrt{n-1} \left( \frac{V_{n-1}}{P_{n-1}} - \boldsymbol{v}_t^{(\ell)}(\boldsymbol{x}_t) \right) \xrightarrow{d} \mathcal{N} \left( 0, \mathbb{E}_{\boldsymbol{x}_0 \sim q} \left[ w(\boldsymbol{x}_0)^2 \left( f(\boldsymbol{x}_0) - \boldsymbol{v}_t^{(\ell)}(\boldsymbol{x}_t) \right)^2 \right] \right).$$

Expanding the variance term above gives us

$$\mathbb{E}_{\boldsymbol{x}_0 \sim q} \left[ w(\boldsymbol{x}_0)^2 \left( f(\boldsymbol{x}_0) - \boldsymbol{v}_t^{(\ell)}(\boldsymbol{x}_t) \right)^2 \right]$$

$$= \mathbb{E}_{\boldsymbol{x}_0 \sim q} \left[ \frac{p_t(\boldsymbol{x}_0 \mid \boldsymbol{x}_t)^2}{q(\boldsymbol{x}_0)^2} \cdot \left( \boldsymbol{v}_t^{(\ell)}(\boldsymbol{x}_t \mid \boldsymbol{x}_0) - \boldsymbol{v}_t^{(\ell)}(\boldsymbol{x}_t) \right)^2 \right]$$

$$= \int \frac{p_t(\boldsymbol{x}_0 \mid \boldsymbol{x}_t)^2}{q(\boldsymbol{x}_0)} \cdot \left( \boldsymbol{v}_t^{(\ell)}(\boldsymbol{x}_t \mid \boldsymbol{x}_0) - \boldsymbol{v}_t^{(\ell)}(\boldsymbol{x}_t) \right)^2 \, \mathrm{d}\boldsymbol{x}_0$$

$$= \frac{1}{p_t(\boldsymbol{x}_t)} \int p_t(\boldsymbol{x}_0 \mid \boldsymbol{x}_t) \cdot \frac{p_t(\boldsymbol{x}_0 \mid \boldsymbol{x}_t) \, p_t(\boldsymbol{x}_t)}{q(\boldsymbol{x}_0)} \left( \boldsymbol{v}_t^{(\ell)}(\boldsymbol{x}_t \mid \boldsymbol{x}_0) - \boldsymbol{v}_t^{(\ell)}(\boldsymbol{x}_t) \right)^2 \, \mathrm{d}\boldsymbol{x}_0$$

$$= \frac{1}{p_t(\boldsymbol{x}_t)} \int p_t(\boldsymbol{x}_0 \mid \boldsymbol{x}_t) \cdot p_t(\boldsymbol{x}_t \mid \boldsymbol{x}_0) \left( \boldsymbol{v}_t^{(\ell)}(\boldsymbol{x}_t \mid \boldsymbol{x}_0) - \boldsymbol{v}_t^{(\ell)}(\boldsymbol{x}_t) \right)^2 \, \mathrm{d}\boldsymbol{x}_0$$

$$= \frac{1}{p_t(\boldsymbol{x}_t)} \cdot \mathbb{E}_{\boldsymbol{x}_0 \sim p_t(\cdot \mid \boldsymbol{x}_t)} \left[ p_t(\boldsymbol{x}_t \mid \boldsymbol{x}_0) \left( \boldsymbol{v}_t^{(\ell)}(\boldsymbol{x}_t \mid \boldsymbol{x}_0) - \boldsymbol{v}_t^{(\ell)}(\boldsymbol{x}_t) \right)^2 \right],$$

as desired. $\square$

**Lemma 4.** *Assume $\boldsymbol{v}_t$ is bounded. Then we have*

$$\mathrm{Var}\left( \widehat{\boldsymbol{v}}_t^{(\ell)} \mid \boldsymbol{x}_t, I = i \right) \approx \frac{1}{n-1} \, \mathbb{E}_{\boldsymbol{x}_0 \sim p_t(\cdot \mid \boldsymbol{x}_t)} \left[ \frac{p_t(\boldsymbol{x}_t \mid \boldsymbol{x}_0)}{p_t(\boldsymbol{x}_t)} \left( \boldsymbol{v}_t^{(\ell)}(\boldsymbol{x}_t \mid \boldsymbol{x}_0) - \boldsymbol{v}_t^{(\ell)}(\boldsymbol{x}_t) \right)^2 \right]$$

*for large $n$.*

*Proof.* Let $g(x, y) = x/y$ so that $\widehat{\boldsymbol{v}}_t^{(\ell)} = g(V_{n-1} + v_i, P_{n-1} + p_i)$. Performing a Taylor expansion of $g$ at the point $(V_{n-1}, P_{n-1})$ gives us

$$\widehat{\boldsymbol{v}}_t^{(\ell)} = g(V_{n-1}, P_{n-1}) + \nabla g(V_{n-1} + \lambda_{n-1} v_i, P_{n-1} + \lambda_{n-1} p_i) \begin{bmatrix} v_i \\ p_i \end{bmatrix}$$

$$= \frac{V_{n-1}}{P_{n-1}} + \frac{v_i}{P_{n-1} + \lambda_{n-1} p_i} - \frac{V_{n-1} + \lambda_{n-1} v_i}{(P_{n-1} + \lambda_{n-1} p_i)^2} \, p_i$$

for some $[0, 1]$-valued random variable $\lambda_{n-1}$. It follows that

$$\sqrt{n-1} \left( \widehat{\boldsymbol{v}}_t^{(\ell)} - \boldsymbol{v}_t^{(\ell)}(\boldsymbol{x}_t) \right)$$

$$= \sqrt{n-1} \left( \frac{V_{n-1}}{P_{n-1}} - \boldsymbol{v}_t^{(\ell)}(\boldsymbol{x}_t) \right) + \frac{\sqrt{n-1} \cdot v_i}{P_{n-1} + \lambda_{n-1} p_i} - \sqrt{n-1} \cdot \frac{V_{n-1} + \lambda_{n-1} v_i}{(P_{n-1} + \lambda_{n-1} p_i)^2} \cdot p_i \quad (29)$$

For the second term in Eq. (29), we first note that by the law of large numbers and Eq. (24), we have $\frac{P_{n-1}}{n-1} \xrightarrow{p} p_t(\boldsymbol{x}_t)$ as $n \to \infty$. Also note that $\frac{\lambda_{n-1}}{n-1} \to 0$ as $n \to \infty$ deterministically. It follows that $\frac{1}{n-1}(P_{n-1} + \lambda_{n-1} p_i) \xrightarrow{p} p_t(\boldsymbol{x}_t)$. We also have $\frac{v_i}{\sqrt{n-1}} \to 0$ deterministically since $\boldsymbol{v}_t$ is bounded. Hence, we have

$$\frac{\sqrt{n-1} \cdot v_i}{P_{n-1} + \lambda_{n-1} p_i} = \frac{\frac{1}{\sqrt{n-1}} v_i}{\frac{1}{n-1}(P_{n-1} + \lambda_{n-1} p_i)} \xrightarrow{p} 0$$

by Slutsky's theorem, so the second term of Eq. (29) converges to 0 in probability.

For the third term in Eq. (29), we have $\frac{1}{(n-1)^2}(P_{n-1} + \lambda_{n-1}p_i)^2 \xrightarrow{p} p_t(\boldsymbol{x}_t)^2$ by Slutsky's theorem. At the same time, by the law of large numbers and Eq. (25), we have $\frac{V_{n-1}}{n-1} \xrightarrow{p} p_t(\boldsymbol{x}_t)\boldsymbol{v}_t^{(\ell)}(\boldsymbol{x}_t)$. Since $\boldsymbol{v}_t$ is bounded, we have $\frac{\lambda_{n-1}v_i}{n-1} \to 0$ deterministically. We also have $\frac{p_i}{\sqrt{n-1}} \to 0$ deterministically. Therefore, we get that

$$\frac{1}{(n-1)^{3/2}}\left(V_{n-1} + \lambda_{n-1}v_i\right)p_i = \frac{p_i}{\sqrt{n-1}} \cdot \left(\frac{V_{n-1}}{n-1} + \frac{\lambda_{n-1}v_i}{n-1}\right)$$
$$\xrightarrow{p} 0 \cdot \left(p_t(\boldsymbol{x}_t)\boldsymbol{v}_t^{(\ell)}(\boldsymbol{x}_t) + 0\right) = 0,$$

by Slutsky's theorem. Then by Slutsky's theorem again, we have

$$\sqrt{n-1} \cdot \frac{V_{n-1} + \lambda_{n-1}v_i}{(P_{n-1} + \lambda_{n-1}p_i)^2} \cdot p_i = \frac{\frac{1}{(n-1)^{3/2}}\left(V_{n-1} + \lambda_{n-1}v_i\right)p_i}{\frac{1}{(n-1)^2}\left(P_{n-1} + \lambda_{n-1}p_i\right)^2} \xrightarrow{p} 0,$$

so the third term in Eq. (29) also converges to 0 in probability.

Therefore, combining Lemma 3 and the results above using Slutsky's theorem, we conclude that

$$\sqrt{n-1}\left(\widehat{\boldsymbol{v}}_t^{(\ell)} - \boldsymbol{v}_t^{(\ell)}(\boldsymbol{x}_t)\right) \xrightarrow{d} \mathcal{N}\left(0, \mathbb{E}_{\boldsymbol{x}_0 \sim p_t(\cdot|\boldsymbol{x}_t)}\left[\frac{p_t(\boldsymbol{x}_t \mid \boldsymbol{x}_0)}{p_t(\boldsymbol{x}_t)}\left(\boldsymbol{v}_t^{(\ell)}(\boldsymbol{x}_t \mid \boldsymbol{x}_0) - \boldsymbol{v}_t^{(\ell)}(\boldsymbol{x}_t)\right)^2\right]\right).$$

This means that $\widehat{\boldsymbol{v}}_t^{(\ell)}$ has an approximated variance

$$\frac{1}{n-1}\mathbb{E}_{\boldsymbol{x}_0 \sim p_t(\cdot|\boldsymbol{x}_t)}\left[\frac{p_t(\boldsymbol{x}_t \mid \boldsymbol{x}_0)}{p_t(\boldsymbol{x}_t)}\left(\boldsymbol{v}_t^{(\ell)}(\boldsymbol{x}_t \mid \boldsymbol{x}_0) - \boldsymbol{v}_t^{(\ell)}(\boldsymbol{x}_t)\right)^2\right],$$

as desired. □

We are now ready to state and prove the main theorem.

**Theorem 3.** *Fix $t \in [0,1]$. Let $\boldsymbol{v}_t$ be bounded. Let $\varepsilon \in (0,1)$. Assume*

$$M := \int_{\{\boldsymbol{x}:p_t(\boldsymbol{x})\leq\varepsilon\}} \mathbb{E}_{\boldsymbol{x}_0 \sim p_t(\cdot|\boldsymbol{x}_t)}\left[\left\|\boldsymbol{v}_t(\boldsymbol{x}_t \mid \boldsymbol{x}_0) - \boldsymbol{v}_t(\boldsymbol{x}_t)\right\|^2\right]\,\mathrm{d}\boldsymbol{x}_t < \infty.$$

*Then, for large enough $n$, we have*

$$\mathcal{V}_{StableVM}(t) \lessapprox \frac{1}{n-1}\left(\frac{1}{\varepsilon} \cdot \mathcal{V}_{CFM}(t) + M\right).$$

*Proof.* Recall from Eq. (11) that we have

$$\mathcal{V}_{\text{StableVM}}(t) = \mathbb{E}\left[\operatorname{Tr}\operatorname{Cov}\left(\widehat{\boldsymbol{v}}_t \mid \boldsymbol{x}_t\right)\right]$$
$$= \mathbb{E}\left[\operatorname{Tr}\left(\mathbb{E}\left[\operatorname{Cov}\left(\widehat{\boldsymbol{v}}_t \mid \boldsymbol{x}_t, I\right) \mid \boldsymbol{x}_t\right]\right)\right]$$
$$= \mathbb{E}\left[\mathbb{E}\left[\operatorname{Tr}\operatorname{Cov}\left(\widehat{\boldsymbol{v}}_t \mid \boldsymbol{x}_t, I\right) \mid \boldsymbol{x}_t\right]\right],$$

where the second line follows from Lemma 2, and the third line follows from the linearity of expectation. Note that the inner expectation is over the random variable $I$, and the outer expectation is over the random variable $\boldsymbol{x}_t$. From Lemma 4, we have

$$\operatorname{Var}\left(\widehat{\boldsymbol{v}}_t^{(\ell)} \mid \boldsymbol{x}_t, I = i\right) \approx \frac{1}{n-1}\mathbb{E}_{\boldsymbol{x}_0 \sim p_t(\cdot|\boldsymbol{x}_t)}\left[\frac{p_t(\boldsymbol{x}_t \mid \boldsymbol{x}_0)}{p_t(\boldsymbol{x}_t)}\left(\boldsymbol{v}_t^{(\ell)}(\boldsymbol{x}_t \mid \boldsymbol{x}_0) - \boldsymbol{v}_t^{(\ell)}(\boldsymbol{x}_t)\right)^2\right].$$

Then

$$\operatorname{Tr}\operatorname{Cov}\left(\widehat{\boldsymbol{v}}_t \mid \boldsymbol{x}_t, I = i\right) = \sum_{\ell=1}^{d} \operatorname{Var}\left(\widehat{\boldsymbol{v}}_t^{(\ell)} \mid \boldsymbol{x}_t, I = i\right)$$

$$\approx \frac{1}{n-1}\sum_{\ell=1}^{d}\mathbb{E}_{\boldsymbol{x}_0 \sim p_t(\cdot|\boldsymbol{x}_t)}\left[\frac{p_t(\boldsymbol{x}_t \mid \boldsymbol{x}_0)}{p_t(\boldsymbol{x}_t)}\left(\boldsymbol{v}_t^{(\ell)}(\boldsymbol{x}_t \mid \boldsymbol{x}_0) - \boldsymbol{v}_t^{(\ell)}(\boldsymbol{x}_t)\right)^2\right]$$

$$= \frac{1}{n-1}\mathbb{E}_{\boldsymbol{x}_0 \sim p_t(\cdot|\boldsymbol{x}_t)}\left[\frac{p_t(\boldsymbol{x}_t \mid \boldsymbol{x}_0)}{p_t(\boldsymbol{x}_t)}\|\boldsymbol{v}_t(\boldsymbol{x}_t \mid \boldsymbol{x}_0) - \boldsymbol{v}_t(\boldsymbol{x}_t)\|^2\right].$$

Since $i$ does not appear in the last line above, we get

$$\mathcal{V}_{\text{StableVM}}(t) \approx \frac{1}{n-1}\mathbb{E}_{\boldsymbol{x}_t \sim p_t}\left[\mathbb{E}_{\boldsymbol{x}_0 \sim p_t(\cdot|\boldsymbol{x}_t)}\left[\frac{p_t(\boldsymbol{x}_t \mid \boldsymbol{x}_0)}{p_t(\boldsymbol{x}_t)}\|\boldsymbol{v}_t(\boldsymbol{x}_t \mid \boldsymbol{x}_0) - \boldsymbol{v}_t(\boldsymbol{x}_t)\|^2\right]\right]$$

$$\leq \frac{1}{n-1}\mathbb{E}_{\boldsymbol{x}_t \sim p_t}\left[\frac{1}{p_t(\boldsymbol{x}_t)}\mathbb{E}_{\boldsymbol{x}_0 \sim p_t(\cdot|\boldsymbol{x}_t)}\left[\|\boldsymbol{v}_t(\boldsymbol{x}_t \mid \boldsymbol{x}_0) - \boldsymbol{v}_t(\boldsymbol{x}_t)\|^2\right]\right],$$

where we have used $p_t(\boldsymbol{x}_t \mid \boldsymbol{x}_0) \leq 1$ in the second line.

Now we fix some $\varepsilon > 0$. Define $P_{>\varepsilon} := \{\boldsymbol{x} \in \mathbb{R}^d : p_t(\boldsymbol{x}) > \varepsilon\}$ and $P_{\leq\varepsilon} := \mathbb{R}^d \setminus P_{>\varepsilon}$. Then we have

$$\mathcal{V}_{\text{StableVM}}(t) \lessgtr \frac{1}{n-1}\int_{\mathbb{R}^d}\frac{\mathbb{E}_{\boldsymbol{x}_0 \sim p_t(\cdot|\boldsymbol{x}_t)}\left[\|\boldsymbol{v}_t(\boldsymbol{x}_t \mid \boldsymbol{x}_0) - \boldsymbol{v}_t(\boldsymbol{x}_t)\|^2\right]}{p_t(\boldsymbol{x}_t)} \cdot p_t(\boldsymbol{x}_t)\, \mathrm{d}\boldsymbol{x}_t$$

$$= \frac{1}{n-1}\int_{P_{>\varepsilon}}\frac{\mathbb{E}_{\boldsymbol{x}_0 \sim p_t(\cdot|\boldsymbol{x}_t)}\left[\|\boldsymbol{v}_t(\boldsymbol{x}_t \mid \boldsymbol{x}_0) - \boldsymbol{v}_t(\boldsymbol{x}_t)\|^2\right]}{p_t(\boldsymbol{x}_t)} \cdot p_t(\boldsymbol{x}_t)\, \mathrm{d}\boldsymbol{x}_t$$

$$+ \frac{1}{n-1}\int_{\leq\varepsilon}\mathbb{E}_{\boldsymbol{x}_0 \sim p_t(\cdot|\boldsymbol{x}_t)}\left[\|\boldsymbol{v}_t(\boldsymbol{x}_t \mid \boldsymbol{x}_0) - \boldsymbol{v}_t(\boldsymbol{x}_t)\|^2\right]\, \mathrm{d}\boldsymbol{x}_t$$

$$\leq \frac{1}{\varepsilon(n-1)}\int_{P_{>\varepsilon}}\mathbb{E}_{\boldsymbol{x}_0 \sim p_t(\cdot|\boldsymbol{x}_t)}\left[\|\boldsymbol{v}_t(\boldsymbol{x}_t \mid \boldsymbol{x}_0) - \boldsymbol{v}_t(\boldsymbol{x}_t)\|^2\right] \cdot p_t(\boldsymbol{x}_t)\, \mathrm{d}\boldsymbol{x}_t + \frac{M}{n-1}$$

$$\leq \frac{1}{\varepsilon(n-1)}\int_{\mathbb{R}^d}\mathbb{E}_{\boldsymbol{x}_0 \sim p_t(\cdot|\boldsymbol{x}_t)}\left[\|\boldsymbol{v}_t(\boldsymbol{x}_t \mid \boldsymbol{x}_0) - \boldsymbol{v}_t(\boldsymbol{x}_t)\|^2\right] \cdot p_t(\boldsymbol{x}_t)\, \mathrm{d}\boldsymbol{x}_t + \frac{M}{n-1}$$

$$= \frac{1}{\varepsilon(n-1)} \cdot \mathbb{E}_{\boldsymbol{x}_t \sim p_t, \boldsymbol{x}_0 \sim p_t(\cdot|\boldsymbol{x}_t)}\left[\|\boldsymbol{v}_t(\boldsymbol{x}_t \mid \boldsymbol{x}_0) - \boldsymbol{v}_t(\boldsymbol{x}_t)\|^2\right] + \frac{M}{n-1}$$

$$= \frac{1}{n-1}\left(\frac{1}{\varepsilon} \cdot \mathcal{V}_{\text{CFM}}(t) + M\right),$$

as desired. $\qquad\qquad\square$

## D.4 SIMULATING THE REVERSE SDE IN LOW-VARIANCE REGIME

Ma et al. (2024) show that the reverse-time SDE (Eq. (5)) with score function $\boldsymbol{s}_t(\boldsymbol{x}) = \nabla_{\boldsymbol{x}} \log p_t(\boldsymbol{x})$ and arbitrary diffusion strength $w_t \geq 0$ yields the correct marginal density $p_t(\boldsymbol{x})$ at each time $t$. Furthermore, as established in Anderson (1982); Ma et al. (2024), if $\boldsymbol{x}_{t_1} \sim p_{t_1}(\boldsymbol{x})$, then the reverse-time solution $\boldsymbol{x}_\tau$ at any $\tau \in [0, t_1]$ is distributed according to the posterior:

$$p_\tau(\boldsymbol{x}_\tau \mid \boldsymbol{x}_{t_1}) = \mathbb{E}_{p_{t_1}(\boldsymbol{x}_0|\boldsymbol{x}_{t_1})}\left[p_\tau(\boldsymbol{x}_\tau \mid \boldsymbol{x}_0, \boldsymbol{x}_{t_1})\right] \approx p_\tau(\boldsymbol{x}_\tau \mid \boldsymbol{x}_0, \boldsymbol{x}_{t_1}). \qquad (30)$$

**Proposition 2.** *Let $x_t \sim \mathcal{N}(\alpha_t x_0, \ \sigma_t^2 \mathbf{I})$ and $x_\tau \sim \mathcal{N}(\alpha_\tau x_0, \ \sigma_\tau^2 \mathbf{I})$, where $\tau < t$, and $x_0 \sim p(x_0)$ is the clean data sample. For any fixed variance parameter $\beta_t^2 \in (0, \sigma_\tau^2)$, define the posterior distribution as*

$$p_\tau^{\alpha_t}(x_\tau \mid x_t, x_0) = \mathcal{N}(k_t x_t + \lambda_t x_0, \ \beta_t^2 \mathbf{I}),$$

*then the coefficients*

$$k_t = \sqrt{\frac{\sigma_\tau^2 - \beta_t^2}{\sigma_t^2}}, \quad \lambda_t = \alpha_\tau - \alpha_t \cdot \sqrt{\frac{\sigma_\tau^2 - \beta_t^2}{\sigma_t^2}}$$

*guarantee that the marginal of $x_\tau$ is $\mathcal{N}(\alpha_\tau x_0, \ \sigma_\tau^2 \mathbf{I})$.*

*Proof.* We begin by expressing $x_t$ using the forward diffusion process:

$$x_t = \alpha_t x_0 + \sigma_t \epsilon, \quad \epsilon \sim \mathcal{N}(0, \mathbf{I}).$$

We define the reverse model as a Gaussian conditional:

$$x_\tau = k_t x_t + \lambda_t x_0 + \eta, \quad \eta \sim \mathcal{N}(0, \beta_t^2 \mathbf{I}).$$

Substituting $x_t$ yields:

$$x_\tau = k_t(\alpha_t x_0 + \sigma_t \epsilon) + \lambda_t x_0 + \eta = (k_t \alpha_t + \lambda_t) x_0 + k_t \sigma_t \epsilon + \eta.$$

Hence, the conditional distribution of $x_\tau$ given $x_0$ is:

$$x_\tau \mid x_0 \sim \mathcal{N}\left((k_t \alpha_t + \lambda_t) x_0, \ (k_t^2 \sigma_t^2 + \beta_t^2)\mathbf{I}\right).$$

To match the desired marginal $x_\tau \sim \mathcal{N}(\alpha_\tau x_0, \ \sigma_\tau^2 \mathbf{I})$, we require:

$$k_t \alpha_t + \lambda_t = \alpha_\tau,$$
$$k_t^2 \sigma_t^2 + \beta_t^2 = \sigma_\tau^2.$$

Solving the second equation above for $k_t$, we obtain:

$$k_t = \sqrt{\frac{\sigma_\tau^2 - \beta_t^2}{\sigma_t^2}}.$$

Substituting into first equation, we get:

$$\lambda_t = \alpha_\tau - \alpha_t \cdot \sqrt{\frac{\sigma_\tau^2 - \beta_t^2}{\sigma_t^2}}.$$

Thus, the choice of $k_t$ and $\lambda_t$ ensures that the conditional distribution of $x_\tau$ is consistent with the marginal. $\square$

Within this low variance area, we also have

$$v_t(x_t) \approx v_t(x_t \mid x_0) = \frac{\sigma_t'}{\sigma_t}(x_t - \alpha_t x_0) + \alpha_t' x_0, \tag{31}$$

Thus, given the velocity field $v_t(x_t)$ and the current state $x_t$, the target $x_0$ can be extracted as:

$$x_0 = \frac{v_t(x_t) - \frac{\sigma_t'}{\sigma_t} x_t}{\alpha_t' - \frac{\sigma_t'}{\sigma_t} \alpha_t} \tag{32}$$

Plugging in this equation into the original expression, thus the posterior distribution with $\boldsymbol{x}_0$ eliminated via $\boldsymbol{v}_t(\boldsymbol{x}_t)$, is given by:

$$p_\tau(\boldsymbol{x}_\tau \mid \boldsymbol{x}_t, \boldsymbol{v}_t(\boldsymbol{x}_t)) = \mathcal{N}\left(\boldsymbol{\mu}_{\tau|t},\ \beta_t^2 \mathbf{I}\right)$$

where the posterior mean is explicitly:

$$\boldsymbol{\mu}_{\tau|t} = \left(\sqrt{\frac{\sigma_\tau^2 - \beta_t^2}{\sigma_t^2}} - \left(\alpha_\tau - \alpha_t \sqrt{\frac{\sigma_\tau^2 - \beta_t^2}{\sigma_t^2}}\right) \cdot \frac{\frac{\sigma_t'}{\sigma_t}}{\alpha_t' - \frac{\sigma_t'}{\sigma_t}\alpha_t}\right)\boldsymbol{x}_t + \left(\alpha_\tau - \alpha_t \sqrt{\frac{\sigma_\tau^2 - \beta_t^2}{\sigma_t^2}}\right) \cdot \frac{\boldsymbol{v}_t(\boldsymbol{x}_t)}{\alpha_t' - \frac{\sigma_t'}{\sigma_t}\alpha_t}$$

Assuming $\alpha_t = 1 - t$ and $\sigma_t = t$, the DDIM-style posterior becomes:

$$p_\tau(\boldsymbol{x}_\tau \mid \boldsymbol{x}_t, \boldsymbol{v}_t(\boldsymbol{x}_t)) = \mathcal{N}\left(\boldsymbol{\mu}_{\tau|t},\ \beta_t^2 \mathbf{I}\right)$$

with mean:

$$\boldsymbol{\mu}_{\tau|t} = \left(\sqrt{\frac{\tau^2 - \beta_t^2}{t^2}} + \left((1-\tau) - (1-t)\sqrt{\frac{\tau^2 - \beta_t^2}{t^2}}\right)\right)\boldsymbol{x}_t - \left((1-\tau) - (1-t)\sqrt{\frac{\tau^2 - \beta_t^2}{t^2}}\right) t \boldsymbol{v}_t(\boldsymbol{x}_t)$$

If we set $\beta_t = 0$, we obtain the deterministic sampler:

$$\boldsymbol{x}_\tau = \boldsymbol{x}_t + (\tau - t)\boldsymbol{v}_t(\boldsymbol{x}_t)$$

### D.5 EXPLICIT PF-ODE SOLUTION IN LOW-VARIANCE REGIME

In *low-variance regime* ($0 \leq t \leq \xi$), the conditional velocity field simplifies as $\boldsymbol{v}_t(\boldsymbol{x}_t) \approx \boldsymbol{v}_t(\boldsymbol{x}_t \mid \boldsymbol{x}_0)$. We can thus derive explicit solutions to the Probability Flow ODE (PF-ODE) under both the stochastic interpolant and VP diffusion frameworks. We consider the **Probability Flow ODE (PF-ODE)** under the stochastic interpolant framework:

$$\frac{d\boldsymbol{x}_t}{dt} = \boldsymbol{v}_t(\boldsymbol{x}_t) \approx \boldsymbol{v}_t(\boldsymbol{x}_t \mid \boldsymbol{x}_0) = \frac{\sigma_t'}{\sigma_t}(\boldsymbol{x}_t - \alpha_t \boldsymbol{x}_0) + \alpha_t' \boldsymbol{x}_0, \tag{33}$$

where $\alpha_t$ and $\sigma_t$ define a stochastic interpolant, and $\boldsymbol{x}_0$ is the data point to be matched.

**Closed-form of the Target $\boldsymbol{x}_0$**  Given the velocity field $\boldsymbol{v}_t(\boldsymbol{x}_t)$ and the current state $\boldsymbol{x}_t$, the target $\boldsymbol{x}_0$ can be extracted as:

$$\boldsymbol{x}_0 = \frac{\boldsymbol{v}_t(\boldsymbol{x}_t) - \frac{\sigma_t'}{\sigma_t}\boldsymbol{x}_t}{\alpha_t' - \frac{\sigma_t'}{\sigma_t}\alpha_t} = \frac{\boldsymbol{v}_t(\boldsymbol{x}_t) - \frac{\sigma_t'}{\sigma_t}\boldsymbol{x}_t}{C_t}, \tag{34}$$

where we define the coefficient

$$C_t := \alpha_t' - \frac{\sigma_t'}{\sigma_t}\alpha_t.$$

**Solving the PF-ODE from $t_1$ to $0 \leq \tau < t_1$**  We aim to integrate the PF-ODE backward in time from a known terminal state $\boldsymbol{x}_{t_1}$. The PF-ODE can be written as:

$$\frac{d\boldsymbol{x}_t}{dt} + a(t)\boldsymbol{x}_t = b(t), \quad \text{where} \quad a(t) = -\frac{\sigma_t'}{\sigma_t}, \quad b(t) = C_t \boldsymbol{x}_0. \tag{35}$$

This is a linear nonhomogeneous first-order ODE. The integrating factor is:

$$\mu(t) = \exp\left(\int a(t)dt\right) = \exp\left(-\int \frac{\sigma_t'}{\sigma_t}dt\right) = \frac{1}{\sigma_t}.$$

Multiplying both sides by $\mu(t)$ yields:

$$\frac{d}{dt}\left(\frac{\boldsymbol{x}_t}{\sigma_t}\right) = \frac{C_t}{\sigma_t}\boldsymbol{x}_0.$$

Integrating both sides from $t_1$ to $\tau < t_1$:

$$\frac{\boldsymbol{x}_\tau}{\sigma_\tau} = \frac{\boldsymbol{x}_{t_1}}{\sigma_{t_1}} + \int_{t_1}^\tau \frac{C(s)}{\sigma_s}ds \cdot \boldsymbol{x}_0, \tag{36}$$

which gives:

$$\boldsymbol{x}_\tau = \sigma_\tau\left(\frac{\boldsymbol{x}_{t_1}}{\sigma_{t_1}} + I(t_1,\tau)\cdot\boldsymbol{x}_0\right), \tag{37}$$

where

$$I(t_1,\tau) := \int_{t_1}^\tau \frac{C(s)}{\sigma_s}ds.$$

**Substituting $\boldsymbol{x}_0$ in Closed Form**    We now substitute the expression for $\boldsymbol{x}_0$ evaluated at time $t_1$:

$$\boldsymbol{x}_0 = \frac{\boldsymbol{v}_{t_1}(\boldsymbol{x}_{t_1}) - \frac{\sigma_{t_1}'}{\sigma_{t_1}}\boldsymbol{x}_{t_1}}{C_{t_1}}.$$

Substitute this into the solution:

$$\boldsymbol{x}_\tau = \sigma_\tau\left(\frac{\boldsymbol{x}_{t_1}}{\sigma_{t_1}} + I(t_1,\tau)\cdot\frac{\boldsymbol{v}_{t_1}(\boldsymbol{x}_{t_1}) - \frac{\sigma_{t_1}'}{\sigma_{t_1}}\boldsymbol{x}_{t_1}}{C_{t_1}}\right) \tag{38}$$

$$= \sigma_\tau\left[\left(\frac{1}{\sigma_{t_1}} - \frac{\sigma_{t_1}'}{\sigma_{t_1}}\cdot\frac{I(t_1,\tau)}{C_{t_1}}\right)\boldsymbol{x}_{t_1} + \frac{I(t_1,\tau)}{C_{t_1}}\boldsymbol{v}_{t_1}(\boldsymbol{x}_{t_1})\right]. \tag{39}$$

**Final Expression (Only in Terms of $\boldsymbol{x}_{t_1}$)**

$$\boxed{\boldsymbol{x}_\tau = \sigma_\tau\left[\left(\frac{1}{\sigma_{t_1}} - \frac{\sigma_{t_1}'}{\sigma_{t_1}}\cdot\frac{I(t_1,\tau)}{C_{t_1}}\right)\boldsymbol{x}_{t_1} + \frac{I(t_1,\tau)}{C_{t_1}}\boldsymbol{v}_{t_1}(\boldsymbol{x}_{t_1})\right]} \tag{40}$$

This provides a fully explicit backward solution to the PF-ODE, depending only on $\boldsymbol{x}_{t_1}$ and the velocity field $\boldsymbol{v}_{t_1}(\boldsymbol{x}_{t_1})$.

**Special Case: Linear Interpolant**    For the linear interpolant with $\alpha_t = 1 - t$ and $\sigma_t = t$, we have:

$$\alpha_t' = -1, \quad \sigma_t' = 1, \quad \Rightarrow \quad C_t = -\frac{1}{t}, \quad \frac{C_t}{\sigma_t} = -\frac{1}{t^2}.$$

Then:

$$I(t_1,\tau) = \int_{t_1}^\tau -\frac{1}{s^2}ds = \frac{1}{\tau} - \frac{1}{t_1}.$$

Also note:

$$\frac{\sigma'_{t_1}}{\sigma_{t_1}} = \frac{1}{t_1}, \quad C_{t_1} = -\frac{1}{t_1}.$$

Plug into the general expression:

$$\boldsymbol{x}_\tau = \tau \left[ \left( \frac{1}{t_1} - \frac{1}{t_1} \cdot \frac{1/\tau - 1/t_1}{-1/t_1} \right) \boldsymbol{x}_{t_1} + \left( \frac{1/\tau - 1/t_1}{-1/t_1} \right) \boldsymbol{v}_{t_1}(\boldsymbol{x}_{t_1}) \right] \tag{41}$$

$$= \tau \left[ \left( \frac{1}{t_1} + \left( \frac{1}{\tau} - \frac{1}{t_1} \right) \right) \boldsymbol{x}_{t_1} + \left( \frac{1}{t_1} - \frac{1}{\tau} \right) \boldsymbol{v}_{t_1}(\boldsymbol{x}_{t_1}) \right] \tag{42}$$

$$= \boldsymbol{x}_{t_1} + (\tau - t_1)\boldsymbol{v}_{t_1}(\boldsymbol{x}_{t_1}). \tag{43}$$

**Final Linear Interpolant Result**

$$\boxed{\boldsymbol{x}_\tau = \boldsymbol{x}_{t_1} + (\tau - t_1)\boldsymbol{v}_{t_1}(\boldsymbol{x}_{t_1})} \tag{44}$$

In the case of the linear interpolant, the PF-ODE corresponds to a straight-line trajectory with constant velocity $\boldsymbol{v}_{t_1}(\boldsymbol{x}_{t_1})$, enabling exact integration via Euler steps of arbitrary size.

# E   EXPERIMENTAL DETAILS& RESULTS

## E.1   EVALUATION ON STABLE VELOCITY MATCHING

### E.1.1   EXPERIMENTAL SETUP

**Training hyperparameters.**   Our implementation builds upon the official REPA codebase (Yu et al., 2024). We adopt SiT-XL (Ma et al., 2024) as the backbone architecture. The model is optimized using AdamW (Kingma, 2015) with a constant learning rate of $1 \times 10^{-4}$, $\beta_1 = 0.9$, $\beta_2 = 0.999$, and no weight decay. To accelerate training, we employ mixed-precision (FP16) arithmetic together with gradient clipping. Additionally, we use an adaptive EMA rate schedule following Lipman et al. (2024a). Specifically, at each training step, we increment a global counter `num_updates` and compute the EMA decay as

$$\text{decay} = \min \left( \beta, \frac{1 + \texttt{num\_updates}}{10 + \texttt{num\_updates}} \right),$$

where $\beta = 0.9999$ is the asymptotic upper bound. This schedule begins with a low decay (approximately 0.1 at step 0), enabling rapid adaptation of the shadow parameters during early training, and gradually increases toward $\beta$ as training progresses. The rising decay places increasing emphasis on historical parameters, thereby stabilizing the EMA model in later stages. We use a fixed batch size of 256 and do not apply any data augmentation.

**Data preprocessing.**   We strictly adhere to the data preprocessing protocol of ADM (Dhariwal & Nichol, 2021). Input images are encoded into latent vectors $\mathbf{z} \in \mathbb{R}^{32 \times 32 \times 4}$ using the pretrained VAE from Stable Diffusion (Rombach et al., 2022).

**Evaluation details.**   Following Dhariwal & Nichol (2021), we use the same reference batches and evaluation setup as in the official ADM implementation.[1] For faster sampling, we enable TF32 precision, consistent with the REPA setup.

We evaluate generation quality using the following standard metrics:

---

[1]https://github.com/openai/guided-diffusion/tree/main/evaluations

Table 3: Two-stage experiment, which compares the SiT framework and our StableVM framework with bank size 256. Each model is trained for 500k steps, and the checkpoints are evaluated every 40k steps.

| # Steps | SiT | | | | | StableVM (Bank=256) | | | | |
|---|---|---|---|---|---|---|---|---|---|---|
| | IS ↑ | FID ↓ | sFID ↓ | Precision ↑ | Recall ↑ | IS ↑ | FID ↓ | sFID ↓ | Precision ↑ | Recall ↑ |
| 40k (SiT) / 50k (StableVM) | 212.5 | 4.33 | 6.29 | 0.71 | 0.69 | **219.2** | **3.92** | **6.04** | **0.72** | 0.69 |
| 80k | 223.8 | 3.53 | 5.61 | 0.73 | **0.68** | 227.3 | 3.35 | 5.61 | 0.73 | 0.67 |
| 120k | 234.4 | 3.07 | **5.34** | 0.74 | 0.67 | 234.8 | 3.02 | 5.40 | 0.74 | 0.67 |
| 160k | **239.7** | **2.81** | **5.24** | 0.74 | **0.67** | 238.9 | 2.87 | 5.74 | 0.74 | 0.66 |
| 200k | 244.7 | 2.62 | **5.20** | 0.75 | 0.67 | **247.0** | **2.59** | 5.21 | 0.75 | 0.67 |
| 240k | 247.4 | 2.54 | **5.17** | 0.75 | 0.67 | 250.3 | 2.48 | 5.20 | 0.75 | 0.67 |
| 280k | 250.9 | 2.47 | 5.14 | 0.75 | 0.66 | 253.3 | 2.40 | 5.13 | 0.75 | **0.67** |
| 320k | 251.7 | 2.39 | 5.17 | 0.75 | **0.67** | 255.4 | 2.34 | 5.13 | 0.75 | 0.66 |
| 360k | 255.4 | 2.35 | 5.16 | 0.75 | **0.67** | 258.5 | 2.28 | 5.11 | **0.76** | 0.66 |
| 400k | 258.6 | 2.28 | 5.17 | 0.76 | 0.67 | 261.7 | 2.17 | **4.98** | 0.76 | 0.67 |
| 440k | 259.3 | 2.29 | 5.20 | 0.76 | 0.67 | 262.3 | 2.16 | **5.00** | 0.76 | 0.67 |
| 480k | 259.7 | 2.26 | 5.22 | 0.76 | **0.67** | 263.1 | 2.16 | **5.03** | 0.76 | 0.66 |
| 500k | 260.1 | 2.26 | 5.23 | 0.76 | 0.67 | **262.7** | **2.17** | **5.06** | 0.76 | 0.67 |

- **FID** (Heusel et al., 2017): Measures the Fréchet distance between feature distributions of real and generated images using the Inception-v3 network (Szegedy et al., 2016), under the assumption that both are multivariate Gaussian.

- **sFID** (Nash et al., 2021): Computes FID using spatially resolved intermediate features from Inception-v3 to better capture local structure and spatial coherence.

- **IS** (Salimans et al., 2016): Evaluates sample diversity and quality via the Inception-v3 logits, defined as the KL divergence between the marginal label distribution and the conditional distribution of predicted labels.

- **Precision and Recall** (Kynkäänniemi et al., 2019): Quantify sample fidelity (precision) and coverage of the true data manifold (recall) based on nearest-neighbor distances in feature space.

### E.1.2  TWO-STAGE TRAINING AND SAMPLING

For the two-stage experiment, we use $\xi = 0.7$ as the split point between the low-variance regime and the high-variance regime. During training, we sample the timestep $t$ uniformly from $[\xi, 1]$ and compute the loss for optimization. We use the CFM loss as the baseline to compare with our StableVM loss. Other configurations is the same as those stated in Appendix E.1.1. In particular, for $t \in [\xi, 1]$, we train an SiT-XL/2 from scratch, and for $t \in [0, \xi]$, we use a pretrained SiT-XL/2 from REPA [2] for stepping. Table 3 shows the detailed performance results of the two methods. Training is done up to 500k steps, and the checkpoints are evaluated every 40k steps. We note from the results that our model consistently perform better than the SiT with CFM loss.

### E.1.3  COMPARISONS WITH BASELINES

We further extend the training duration to 2M iterations and compare our SiT-XL model trained with the CFM and StableVM loss against a range of strong baselines: ADM (Dhariwal & Nichol, 2021), VDM++ (Kingma & Gao, 2024), Simple Diffusion (Hoogeboom et al., 2023), CDM (Ho et al., 2022), LDM (Rombach et al., 2022), U-ViT (Bao et al., 2023), DiffiT (Hatamizadeh et al., 2024), MDTv2 (Gao

---
[2]https://github.com/sihyun-yu/REPA/blob/main/utils.py#14

Table 4: Comparisons on ImageNet $256 \times 256$ with CFG. $\downarrow$ and $\uparrow$ indicate whether lower or higher values are better, respectively.

| Model | FID$\downarrow$ | sFID$\downarrow$ | IS$\uparrow$ | Precision$\uparrow$ | Recall$\uparrow$ |
|---|---|---|---|---|---|
| *Pixel diffusion* | | | | | |
| ADM-U | 3.94 | 6.14 | 186.7 | 0.82 | 0.52 |
| VDM++ | 2.40 | - | 225.3 | - | - |
| Simple diffusion | 2.77 | - | 211.8 | - | - |
| CDM | 4.88 | - | 158.7 | - | - |
| *Latent diffusion, U-Net* | | | | | |
| LDM-4 | 3.60 | - | 247.7 | 0.87 | 0.48 |
| *Latent diffusion, Transformer + U-Net hybrid* | | | | | |
| U-ViT-H/2 | 2.29 | 5.68 | 263.9 | 0.82 | 0.57 |
| DiffiT* | 1.73 | - | 276.5 | 0.80 | 0.62 |
| MDTv2-XL/2* | 1.58 | 4.52 | 314.7 | 0.79 | 0.65 |
| *Latent diffusion, Transformer* | | | | | |
| MaskDiT | 2.28 | 5.67 | 276.6 | 0.80 | 0.61 |
| SD-DiT | 3.23 | - | - | - | - |
| DiT-XL/2 | 2.27 | 4.60 | 278.2 | **0.83** | 0.57 |
| SiT-XL/2 | 2.05 | 4.46 | 255.4 | 0.81 | 0.59 |
| + StableVM(bank=256) | 2.03 | 4.40 | 260.1 | 0.82 | 0.59 |

et al., 2023), MaskDiT (Zheng et al., 2024), DiT (Peebles & Xie, 2023), and SiT (Ma et al., 2024). Full results are reported in Table 4. Despite training for only 2M iterations—significantly fewer than the 7M iterations used in Ma et al. (2024)—our model achieves competitive performance and closely matches the results reported for SiT. Moreover, under this training budget, our approach converges faster and attains better performance than the CFM-trained SiT baseline.

### E.1.4 UNCONDITIONAL GENERATION

We also evaluate Algorithm 1 in the unconditional generation setting. Specifically, we construct a synthetic Gaussian Mixture Model (GMM) distribution and train the model to learn it using either the standard CFM loss or our proposed StableVM loss.

The GMM is defined with 100 modes. For each component $k$, the mean vector $\boldsymbol{\mu}_k$ is sampled independently from a uniform distribution over $[-1, 1]$ in each dimension. The variances are drawn independently per component and per dimension from a uniform distribution over $[10^{-2}, 10^{-1}]$, yielding anisotropic Gaussian components. The mixing coefficients $\boldsymbol{\pi}$ are obtained by sampling each entry from $\text{Uniform}(0.1, 1.0)$ and normalizing so that they sum to one. To generate samples, we first draw a component index $k$ via multinomial sampling according to $\boldsymbol{\pi}$, then sample from the corresponding Gaussian using the reparameterization trick:

$$\boldsymbol{x} = \boldsymbol{\mu}_k + \boldsymbol{\sigma}_k \odot \boldsymbol{\epsilon}, \quad \text{where} \quad \boldsymbol{\epsilon} \sim \mathcal{N}(0, \mathbf{I}),$$

and $\odot$ denotes element-wise multiplication.

We fix the data dimensionality to 10 and evaluate both the CFM loss and our proposed StableVM loss with a reference batch size of 2048 on this distribution. Model performance is assessed by computing the *second-order moment* of the discrepancy between the model's predicted velocity field $\boldsymbol{v}_{\boldsymbol{\theta}}(\boldsymbol{x}_t, t)$ and the true velocity

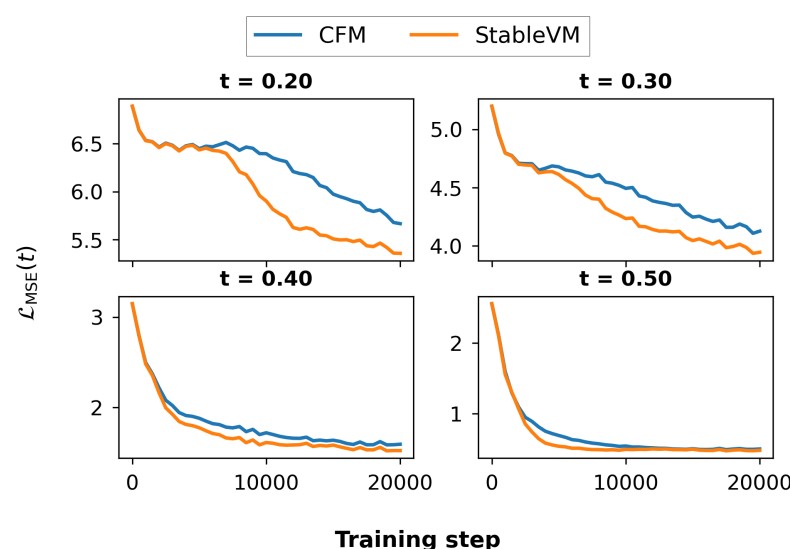

Figure 5: Comparison of StableVM and CFM on the synthetic GMM distribution. We plot the *second-order moment* as a function of training iterations at four time steps: $t = 0.20, 0.30, 0.40$, and $0.50$.

field $\boldsymbol{v}_t(\boldsymbol{x}_t)$, under the marginal distribution $p_t(\boldsymbol{x}_t)$:

$$\mathcal{L}_{\text{MSE}}(t) \;=\; \tfrac{1}{2}\, \mathbb{E}_{\boldsymbol{x}_t \sim p_t}\left[\left\|\boldsymbol{v_\theta}(\boldsymbol{x}_t, t) - \boldsymbol{v}_t(\boldsymbol{x}_t)\right\|^2\right]. \tag{45}$$

However, the exact velocity field $\boldsymbol{v}_t(\boldsymbol{x}_t) = \mathbb{E}_{p_t(\boldsymbol{x}_0|\boldsymbol{x}_t)}[\boldsymbol{v}_t(\boldsymbol{x}_t \mid \boldsymbol{x}_0)]$ is intractable. We therefore approximate it using a self-normalized importance sampling estimator (Hesterberg, 1995). Concretely, given $N = 50{,}000$ samples $\{\boldsymbol{x}_0^i\}_{i=1}^N$ drawn from the data distribution, we approximate:

$$\boldsymbol{v}_t(\boldsymbol{x}_t) \approx \sum_{i=1}^N w_i(\boldsymbol{x}_t)\, \boldsymbol{v}_t(\boldsymbol{x}_t \mid \boldsymbol{x}_0^i), \quad \text{where} \quad w_i(\boldsymbol{x}_t) = \frac{p_t(\boldsymbol{x}_t \mid \boldsymbol{x}_0^i)}{\sum_{j=1}^N p_t(\boldsymbol{x}_t \mid \boldsymbol{x}_0^j)}.$$

This approximation enables us to estimate the *second-order moment* at any time step $t$.

The results are shown in Fig. 5, with separate curves for $t = 0.20, 0.30, 0.40$, and $0.50$. The plots clearly demonstrate that StableVM improves both the convergence speed and the final performance compared to standard CFM.

**Reproduction of Fig. 1.** To reproduce the variance curves in Fig. 1, we follow the same procedure described above but increase the dimensionality of the synthetic GMM to 100 and 500, using identical configuration settings. The true velocity field is again approximated using the self-normalized importance estimator with $N = 10{,}000$ samples. For the CIFAR-10 curve, we use the full training set of 50,000 samples for evaluation.

Table 5: Evaluation results of Wan2.2 on video generation with 25 steps. Higher is better ($\uparrow$) except for LPIPS ($\downarrow$).

| Method | PSNR $\uparrow$ | SSIM $\uparrow$ | LPIPS $\downarrow$ |
|---|---|---|---|
| 25 steps(Ours) | 28.874 | 0.907 | 0.066 |
| 25 steps | 12.379 | 0.477 | 0.551 |

## E.2 STABLE VELOCITY SAMPLING

### E.2.1 EXPERIMENTAL SETUP

We implement our algorithm primarily using the Huggingface `diffusers` library (von Platen et al.), which supports state-of-the-art pretrained models. Following the recommended configurations, we adopt guidance scales of 3.5, 5.0, and 4.5 for Flux (Labs, 2024), SD3 (Esser et al., 2024), and SD3.5 (Esser et al., 2024), respectively. For Wan2.2 (Wan et al., 2025), we set the `guidance scale` to 4.0 and `guidance scale 2` to 3.0.

Our algorithm introduces three key hyperparameters: the split point $\xi$, the number of steps in the low-variance regime $\text{NFE}_{\text{Low}}$, and the variance factor $f_\beta$. As discussed in Sec. 3.2 and Eq. (12): (i) the split point $\xi$ determines the boundary between the *low-variance* and *high-variance* regimes, (ii) the variance factor $f_\beta$ controls the variance of the DDIM-style posterior in Eq. (12), and (iii) $\text{NFE}_{\text{Low}}$ specifies the number of sampling steps allocated to the *low-variance regime*. In practice, we partition the default time scheduler of pretrained models into two regimes according to $\xi$. The low-variance regime is then uniformly divided into $\text{NFE}_{\text{Low}}$ steps, and sampling is performed following Eq. (12). For Tab. 2, we set $\xi = 0.85$, NFELow $= 8$, and $f_\beta = 0$.

**Evaluation Details.** For text-to-image tasks, we evaluate different samplers on the *Geneval* benchmark (Ghosh et al., 2023), which consists of 553 prompts spanning six categories. For each experiment, we generate 4 samples using random seeds $\{0, 1000, 2000, 3000\}$, strictly following the official *Geneval* evaluation protocol.

For text-to-video tasks, we employ Wan2.2 to generate videos conditioned on 100 prompts from the `human_action` dimension of VBench (Huang et al., 2024). We report only reference-based metrics by comparing generated videos against 50-step baseline results. All generations are performed with the same random seed for consistency. Table 5 presents the evaluation results of Wan2.2 on video generation with 25 steps. Our custom sampler achieves significantly higher PSNR and SSIM and much lower LPIPS, indicating that it produces results much closer to the 50-step baseline compared to the original sampler.

### E.2.2 ABLATION STUDIES

Table 6 presents the effect of three key hyperparameters—variance factor $f_\beta$, the number of low-variance steps $\text{NFE}_{\text{Low}}$, and the split point $\xi$—on Stable Velocity Sampling with SD3.5 at $1024 \times 1024$.

Compared to the baseline, introducing a split at $\xi = 0.85$ with $\text{NFE}_{\text{Low}} = 8$ already leads to large gains across all metrics (PSNR from 15.58 to 30.95, LPIPS from 0.370 to 0.050), showing the benefit of separating low- and high-variance regimes.

Varying the variance factor $f_\beta$ reveals a trade-off: increasing $f_\beta$ slightly reduces PSNR but improves overall stability, with $f_\beta = 0.2$ achieving the best overall score (0.722).

Table 6: Ablation study on split point $\xi$, low-variance regime steps $\text{NFE}_{\text{Low}}$, and variance factor $f_\beta$ of Stable Velocity Sampling for SD3.5 at $1024 \times 1024$. Higher is better ($\uparrow$), except for LPIPS ($\downarrow$).

| NFE | $\xi$ | $\text{NFE}_{\text{Low}}$ | $f_\beta$ | Overall $\uparrow$ | PSNR $\uparrow$ | SSIM $\uparrow$ | LPIPS $\downarrow$ |
|---|---|---|---|---|---|---|---|
| **Baseline** | | | | | | | |
| 25 | – | – | – | 0.711 | 15.576 | 0.735 | 0.370 |
| 25 | 0.85 | 8 | 0.0 | 0.718 | 30.954 | 0.958 | 0.050 |
| **Variance factor** $f_\beta$ | | | | | | | |
| 25 | 0.85 | 8 | 0.1 | 0.719 | 30.094 | 0.945 | 0.056 |
| 25 | 0.85 | 8 | 0.2 | **0.722** | 28.536 | 0.915 | 0.071 |
| **Low-variance regime steps** $\text{NFE}_{\text{Low}}$ | | | | | | | |
| 20 | 0.85 | 3 | 0.0 | 0.711 | 26.220 | 0.873 | 0.155 |
| 30 | 0.85 | 13 | 0.0 | 0.719 | 33.891 | 0.977 | 0.025 |
| **Split point** $\xi$ | | | | | | | |
| 20 | 0.90 | 8 | 0.0 | 0.727 | 26.171 | 0.918 | 0.098 |
| 29 | 0.80 | 8 | 0.0 | 0.721 | 34.387 | 0.974 | 0.031 |
| 35 | 0.70 | 8 | 0.0 | 0.721 | **39.303** | **0.987** | **0.014** |

For the number of low-variance steps, allocating more steps consistently improves fidelity: with $\text{NFE}_{\text{Low}} = 13$, PSNR rises to 33.89 and LPIPS drops to 0.025.

Finally, the split point $\xi$ plays a critical role. An earlier split at $\xi = 0.70$ with 35 steps achieves the strongest results overall, reaching the highest PSNR (39.30), SSIM (0.987), and lowest LPIPS (0.014). This confirms that $\xi$ enables the sampler to fully exploit low-variance dynamics for both fidelity and perceptual quality.

# F  LLM USAGE STATEMENT

In preparing this manuscript, we used OpenAI's ChatGPT solely as a language editing tool to polish grammar, phrasing, and clarity of our writing. The model was not involved in research ideation, experimental design, analysis, or drawing conclusions. All scientific content and interpretations are the authors' own.

# G  MORE QUALITATIVE RESULTS

We provide additional qualitative results for SD3.5 (Esser et al., 2024), Flux (Labs, 2024), SD3 (Esser et al., 2024), and Wan2.2 (Wan et al., 2025), shown in Fig. 6, Fig. 7, Fig. 8, and Fig. 9, respectively.

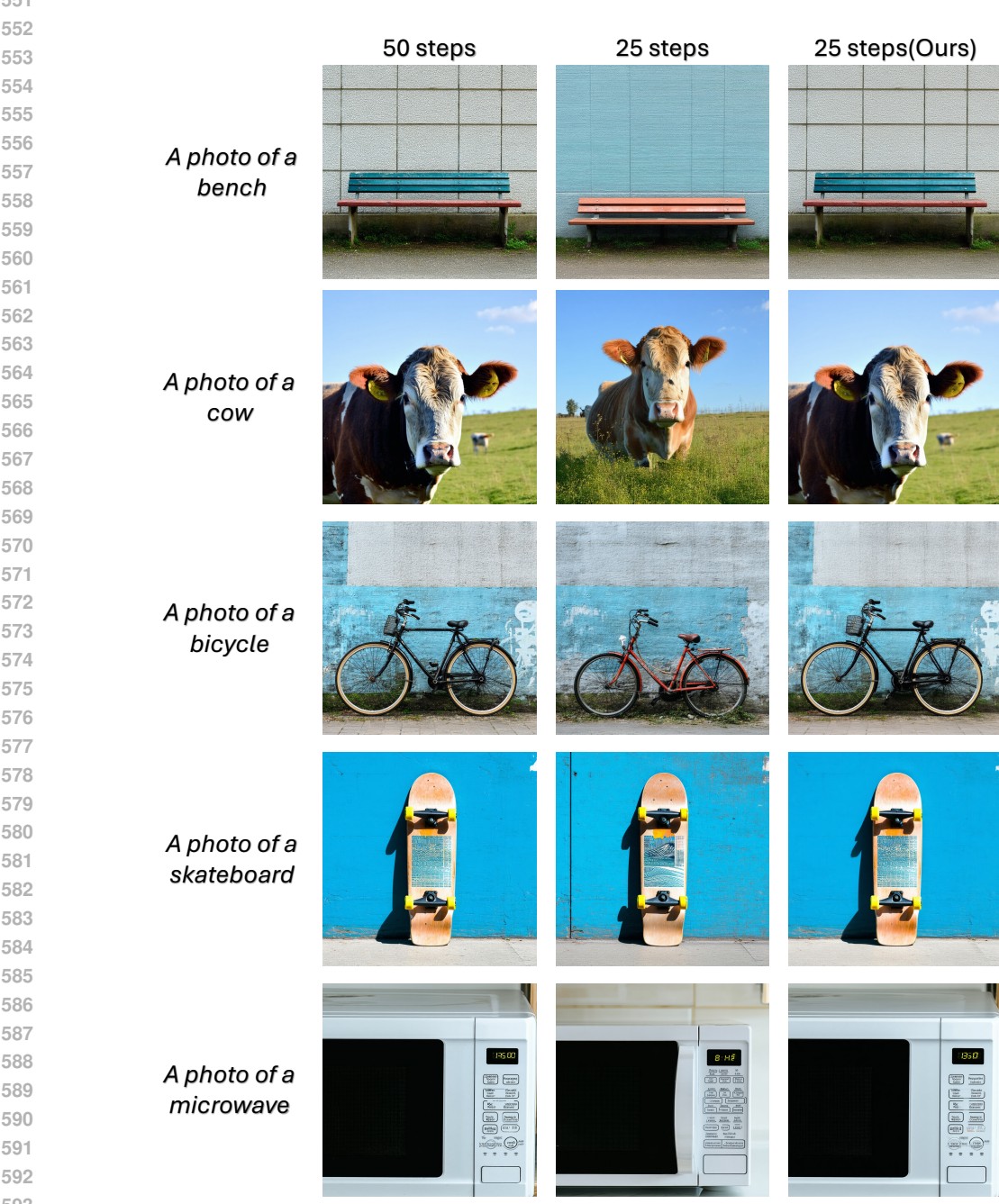

Figure 6: Visual comparison of SD3.5-Large (Esser et al., 2024) on different prompts. We show results using the default sampler with 50 and 25 steps, alongside our 25-step sampler under the same random seed. Zoom in for finer details.

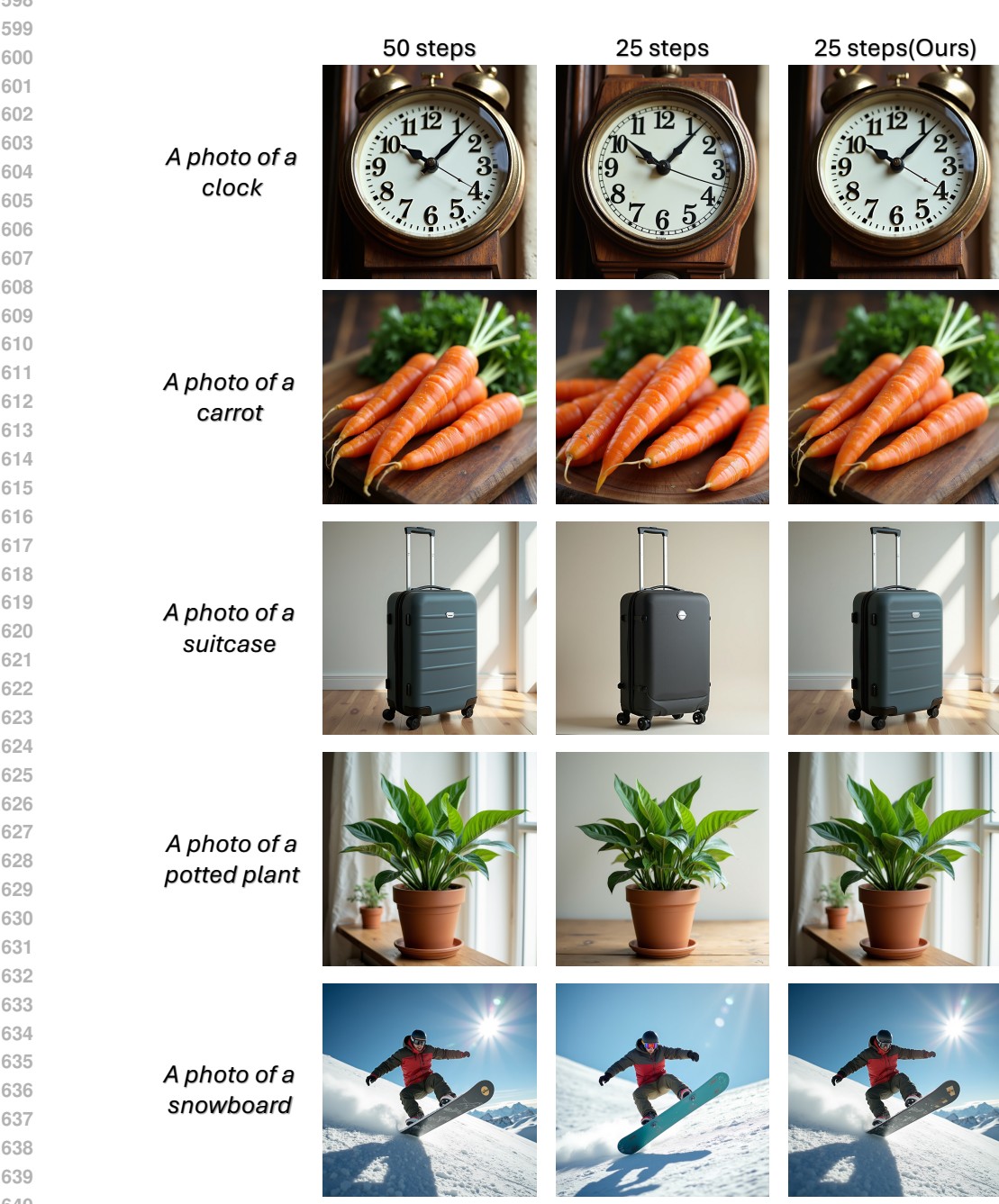

Figure 7: Visual comparison of Flux-dev (Labs, 2024) on different prompts. We show results using the default sampler with 50 and 25 steps, alongside our 25-step sampler under the same random seed. Zoom in for finer details.

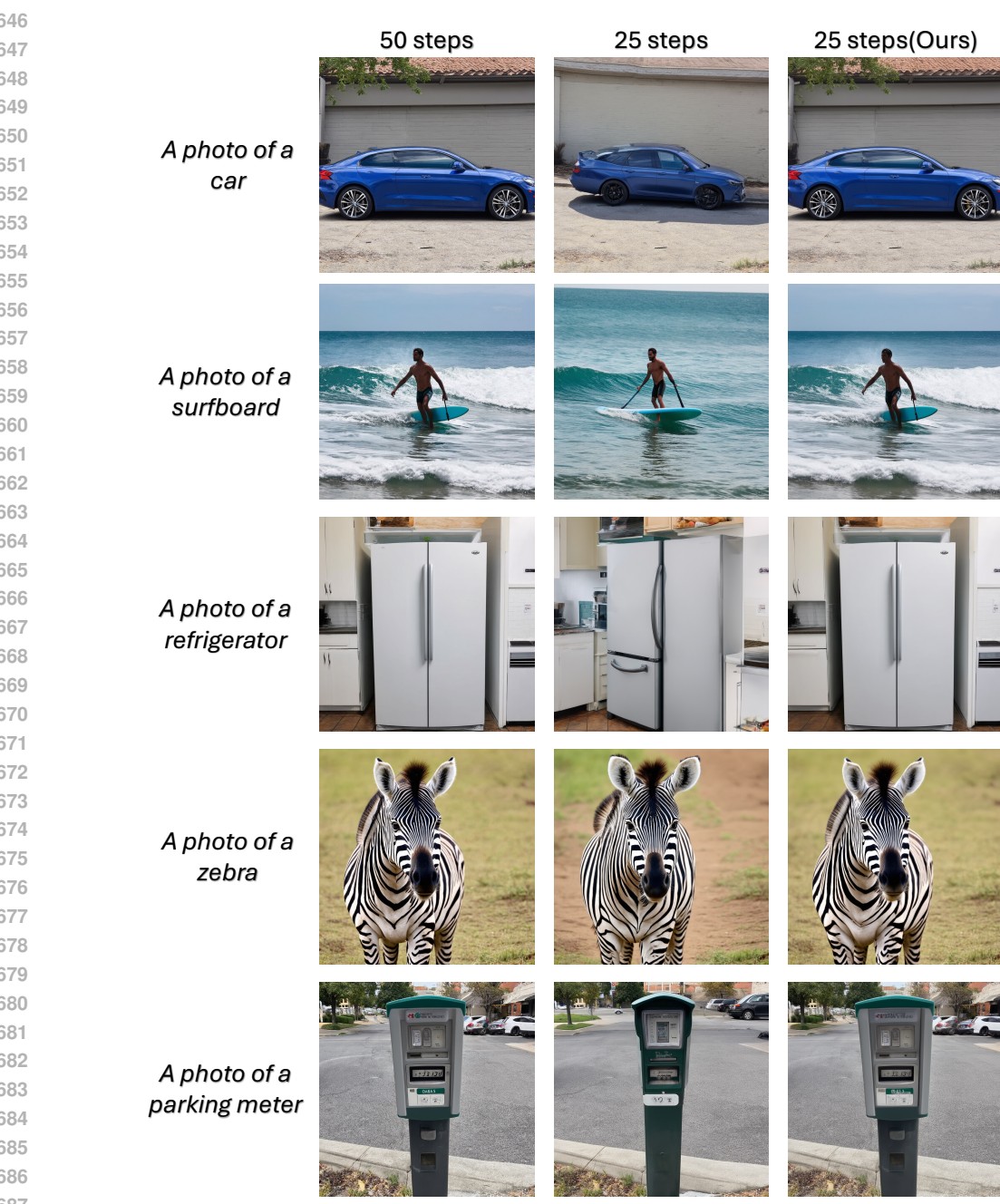

Figure 8: Visual comparison of SD3-medium (Esser et al., 2024) on different prompts. We show results using the default sampler with 50 and 25 steps, alongside our 25-step sampler under the same random seed. Zoom in for finer details.

Prompt: A person is air drumming.

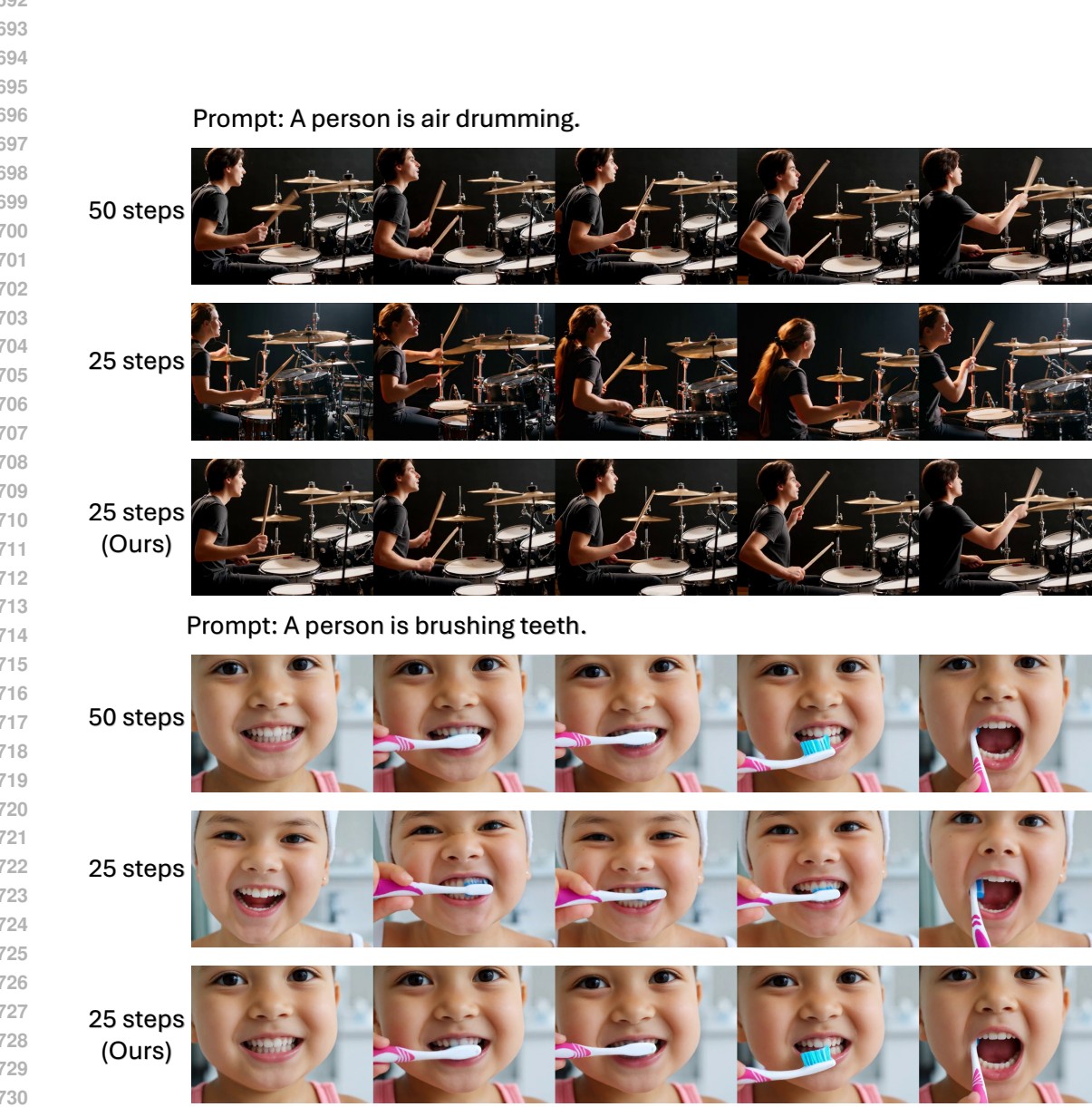

Figure 9: Visual comparison of Wan2.2 (Wan et al., 2025) on different prompts. We show results using the default sampler with 50 and 25 steps, alongside our 25-step sampler under the same random seed. Zoom in for finer details.

