# OpenReview forum: "Fast to Train, Fast to Sample: Stable Velocity for Flow Matching"
_ICLR.cc/2026/Conference — Submitted to ICLR 2026_

### Official Review · Reviewer_HXfF · 2025-10-26

**Soundness:** 2
**Presentation:** 3
**Contribution:** 2
**Rating:** 2
**Confidence:** 3

**Summary:**

This paper aims to improve both the training stability and sampling efficiency of flow matching models. The authors propose decomposing the training process into two distinct regimes: a high-variance regime (corresponding to large injected noise) and a low-variance regime (corresponding to small injected noise).

To stabilize training in the high-variance regime, the paper introduces Stable Velocity Matching, which approximates the true velocity by sampling multiple data points. To accelerate inference in the low-variance regime, it proposes Stable Velocity Sampling, which treats the generated data as the ground truth, effectively linearizing the trajectory into a straight line for faster sampling. The paper provides experimental results for both unconditional and text-to-image generation to validate the proposed methods.

**Strengths:**

- Clarity of Exposition: The paper is well-written and logically structured. Figure 2, in particular, is highly effective. It intuitively illustrates the core differences between high-variance regime and low-variance regime, allowing readers to quickly grasp the paper's key insights.

- Experimental Validation: The paper's key insight—the two-regime decomposition—is clearly supported by Figure 1. The experimental results also demonstrate that the proposed method improves generation quality.

**Weaknesses:**

- Limited Novelty: The paper's contributions appear derivative. Stable Velocity Matching is analogous to using a stable target [1] , and Stable Velocity Sampling strongly resembles DDIM or other straight-line samplers. Since diffusion models are a special case of flow matching, the method seems to be a straightforward adaptation and combination of established diffusion techniques, lacking fundamental novelty.

- Lack of comparison with baselines. The paper lacks a comparison against current state-of-the-art (SOTA) method for efficient and high quality generation. In fact, the proposed method is only compared with variances of itself. To properly contextualize its contributions, the method must be benchmarked against SOTA approaches in terms of both generation quality and efficiency.

Reference: [1] Yilun Xu, Shangyuan Tong, and Tommi Jaakkola. Stable target field for reduced variance score estimation in diffusion models

**Questions:**

Please see weaknesses.

---

> ### Author Response · Authors · 2025-11-20
> **Response to Reviewer HXfF**
>
> Thank you for the insightful and constructive comments. We appreciate your advice and address the questions below.
>
> **Q1: Limited Novelty: StableVM is analogous to STF and StableVS strongly resembles DDIM or other straight-line samplers.**
>
> **A1:** We respectfully disagree with the “limited novelty“ assessment and clarify our contributions below.
>
> Our work originates from a new analysis of the variance behavior in flow matching and stochastic interpolants. We identify a **previously unreported two-regime variance structure**, which has conceptual and algorithmic implications and, to our knowledge, has not been discussed in prior flow-matching literature. While flow matching and diffusion models are fundamentally similar, the dynamics of the velocity field and the score function are still different. In particular, as explored in STF, the variance of the score exhibits three phases, which is different from our two-regime variance structure.
>
> Regarding **StableVM**, although its form superficially resembles that of STF, the underlying mechanism is fundamentally different. As explained in our response to Reviewer kXZ3 Q2, STF operates on a conditional distribution on the first sample of the reference batch, whereas StableVM introduces a **composite conditional distribution** $p_t^{\mathrm{GMM}}(x_t \mid (x_0^i)_{i=1}^n)$ over the whole reference batch. This reformulation **substantially changes the statistical properties of the training target**: the estimator becomes **unbiased** (Theorem 1) and enjoys **provable variance reduction** (Theorems 2 & 3). In contrast, the STF target is biased. Beyond theory, we also extend StableVM to **class-conditional generation**, which, to our knowledge, has not been explored in the STF literature.
>
> For **StableVS**, the connection to DDIM or straight-line samplers arises **only after** a technical simplification that is valid **exclusively in the low-variance regime** identified by our variance analysis. In this regime, the conditional velocity becomes dominated by a single reference sample, which allows the FM dynamics to collapse to a **linear ODE or non-Markovian posterior**. This analytical reduction is what reveals the DDIM-like or PF-ODE-like forms—*not* an assumption we begin with, but a consequence of our two-regime analysis. This insight is absent from existing straight-line or DDIM-based solvers.
>
> Finally, we emphasize that **StableVS is not a standalone sampler** but the low-variance component of a **hybrid sampling strategy**. It must be paired with any high-variance-regime solver (Euler, Heun, DPM, etc.), and we provide a seamless transition mechanism between regimes. Existing straight-line samplers do not offer this regime-aware hybrid design.
>
> In summary, both StableVM and StableVS derive from new variance-based insights that lead to new formulations, new theoretical properties, and new algorithmic behavior that are not present in STF, DDIM, or prior straight-line methods.
>
> **Q2: Lack of comparison with baselines.**
>
> **A2:** We respectfully clarify that comprehensive baseline comparisons are already provided in **Table 4 of Appendix E.1.3**. As further updated in the revision, **StableVM with a larger memory bank (K = 1024)** achieves a best FID of **1.94** at 2M iterations, demonstrating a clear improvement over the standard CFM target.
>
> In addition, our **unconditional synthetic generation experiments** (Appendix E.1.4) provide a controlled setting where StableVM can be directly compared with STF. As shown in our response to Reviewer kXZ3 Q3, StableVM consistently outperforms both CFM and the adapted STF throughout training.
>
> Finally, we also evaluate **StableVS** under various samplers. StableVS is shown to be fully compatible with **Euler-based solvers** (as used in SD3, SD3.5, and Flux.1 dev), the **UniPC sampler** [1] (as used in Wan 2.2), and also with **DPM solvers** [2–3] (see Reviewer 7N2v Q1). These results further substantiate our variance-based analysis and demonstrate that StableVS complements strong existing samplers rather than competing with them.
>
> **References:**
>
> [1] Zhao, Wenliang, et al. “UniPC: A unified predictor–corrector framework for fast sampling of diffusion models.” *Advances in Neural Information Processing Systems* 36 (2023): 49842–49869.
>
> [2] Lu, Cheng, et al. "Dpm-solver++: Fast solver for guided sampling of diffusion probabilistic models." *Machine Intelligence Research* (2025): 1-22.
>
> [3] Xie, Enze, et al. "Sana: Efficient high-resolution image synthesis with linear diffusion transformers." *arXiv preprint arXiv:2410.10629* (2024).

---

### Official Review · Reviewer_kXZ3 · 2025-10-29

**Soundness:** 2
**Presentation:** 3
**Contribution:** 2
**Rating:** 2
**Confidence:** 3

**Summary:**

This paper revisits flow matching from a variance-centric perspective. Motivated from the large variance issue of vanilla conditional flow matching, the authors propose Stable Velocity Matching, a variance-reduced objective that preserves CFM’s global optima while provably improving stability. Extensive experiments are conducted to verify the efficiency of the proposed method.

**Strengths:**

The paper is well organized, starting from observation of large variance in vanilla CFM and then introducing variance reduction techniques. Theotical analysis is also provided to show the benefits of proposed method. The authors conducted extensive experiments to demonstrate the acceleration over vanilla FM.

**Weaknesses:**

1. The motivation in Sec 2 is not very clear to me. I understand that the conditional velocity may have large variance but it doesn't necessarily undermine training stability because the Monte Carlo estimator is to estimate the gradient instead of loss itself. Therefore I suggest the authors compute the variance in gradient using conditional velocity.

2. I don't understand the difference between the proposed method and STF as mentioned in Appendix B. For STF, it first samples a batch $(x_0^i)\_{1 \leq i \leq n}$ from data distribution, and then samples $x_t$ by applying the transition kernel to the "first" training data $x_0^1$. For the proposed method, it first samples $(x_0^i)\_{1\leq i\leq n}$, and theh samples $x_t$ from the mixture model, which is equivalent to choosing a random index $i$ and applying transition kernel to $x_0^i$. Since the order of $(x_0^i)$ doesn't matter, I don't see the differene between the two methods. Could the authors explain more about this?

3. For experiments, the authors only compared the proposed method with vanilla FM and didn't mention even most related methods like STF. If the authors want to claim the advantages or differences of the proposed method over STF, I think a numerical comparison is necessary.

**Questions:**

Please see weakness part.

---

> ### Author Response · Authors · 2025-11-20
> **Response to Reviewer kXZ3 (Part 1/2)**
>
> Thank you for the insightful and constructive comments. We appreciate your advice and address the questions below.
>
> **Q1: The conditional velocity may have large variance but it doesn't necessarily undermine training stability because the Monte Carlo estimator is to estimate the gradient instead of loss itself. Compute the variance in gradient using conditional velocity instead.**
>
> **A1:**  It is true that at a first glance, the variance of the conditional velocity does not necessarily entail anything about training. However, we argue that the large variance of the conditional velocity will also lead to unstable training. We first recall that the CFM loss is
>
> $$
> \mathcal{L}_{\text{CFM}}(\theta,t) =\mathbb{E}\_{q(X_0),p_t(X_t\mid X_0)} \lVert v\_{\theta}(X_t,t)-v_t(X_t\mid X_0) \rVert^2 =\mathbb{E}\_{p_t(X_t)}\mathbb{E}\_{p_t(X_0\mid X_t)} \lVert v\_{\theta}(X_t,t)-v_t(X_t\mid X_0)\rVert^2.
> $$
>
> For clarity, we use upper-case letters to denote random variables and lower-case letters to denote samples. First note that during a stochastic gradient descent training, we sample $x_0$ and $\varepsilon$ from their joint distribution and regress over $v_t(x_t\mid x_0)$. That is, for each training step, we are actually regressing over a Monte-Carlo sample of the estimator $v_t(X_t\mid X_0)$, whose conditional expectation on $X_t=x_t$ is the true velocity field $v_t(x_t)$. So we can view SGD as repeatedly updating the model parameters using Monte-Carlo samples of the true velocity field. This means that if the estimator $v_t(X_t\mid X_0)$ has a large variance, then each step we are regressing over very different targets, which leads to unstable training. Figure 2 in our paper also explains this problem visually. Furthermore, as in Eq. (8) in our paper, the minimizer of the CFM objective for each $x_t$ is $v_{\theta}^*(x_t,t) = \mathbb{E}_{p_t(X_0\mid X_t=x_t)}[v_t(X_t\mid X_0)]$, which is the expectation of the true velocity estimator. Therefore, we can intuitively understand optimizing the CFM loss as sampling from the true velocity estimator. In this case, the variance of the estimator definitely plays an important role in training stability.
>
> We can indeed also compute the variance of the gradient of the CFM loss. The gradient estimator is
>
> $$
> \hat{g}:=\hat{g}(\theta,X_0,X_t)=2J_{\theta}(X_t)^\top\big(v_{\theta}(X_t,t)-v_t(X_t\mid X_0)\big),\quad\text{where }J_{\theta}(X_t)=\frac{\partial v_{\theta}(X_t,t)}{\partial\theta}.
> $$
>
> Conditional on $X_t=x_t$, we have
>
> $$
> \mathrm{Cov} (\hat{g}\mid X_t=x_t) = 4J_{\theta}(x_t)^\top\mathrm{Cov}\big(v_t(x_t\mid X_0)\big)J_{\theta}(x_t).
> $$
>
> We observe that the variance of the conditional velocity also appears in the variance of the gradient. In fact, the rest of the parts in the variance of the gradient is due to the neural network architecture. Therefore, the variance of the conditional velocity is exactly the architecture-agnostic part that reveals the intrinsic noisiness of the CFM training framework.
>
> In conclusion, the variance of the conditional velocity directly affects the stability of training, and our StableVM target stabilizes training with our lower-variance target.

---

> ### Author Response · Authors · 2025-11-20
> **Response to Reviewer kXZ3 (Part 2/2)**
>
> **Q2: The difference between STF and proposed method StableVM.**
>
> **A2:** The key difference between STF and our proposed StableVM lies in the **composite conditional distribution** $p_t^{\text{GMM}}$*.* While both methods draw a reference batch $(x_0^i\)_{i=1}^n$, **the resulting distribution of $x_t$ is fundamentally different**. Importantly, the order of the samples in the reference batch is **fixed** after sampling. STF generates $x_t$ solely by applying the transition kernel to a *single* (the first) sample $x_0^1$, leading to targets that depend heavily on that specific draw. In contrast, StableVM samples $x_t$ from a predefined mixture model over the entire reference set. This composite mechanism covers many more modes and removes the bias arising from relying on a single reference point. Despite the fact that our target and the STF target has the same form, the random variables follow different distributions in our target and the STF target. Because of this reformulation, our target is an unbiased estimator of the true velocity field (Theorem 1) as opposed to the biased STF target. The new formulation also requires a re-derivation of the variance bound (Theorems 2 & 3).
>
> Beyond this core contribution, our framework is the first work to systematically analyze the **variance structure of stochastic interpolants or flow matching**. While flow matching and diffusion models are fundamentally similar, the velocity field and the score function has different dynamics. Unlike the score function in diffusion models, which exhibit a three-phase variance pattern with a high-variance middle regime, the velocity in stochastic interpolants display a **two-regime** structure. This insight motivates our training-free sampling strategy, StableVS, which exploits the low-variance regime for acceleration.
>
> Finally, in our framework, we also extend to **class-conditional generation** via per-class memory banks.
>
> In summary, we (1) debias and generalize STF through a composite conditional formulation, (2) provide the first variance-centric analysis of stochastic interpolants, thereby bridging training- and sampling-acceleration strategies within a unified perspective, and (3) extend it to class-conditional generation.
>
> **Q3: Numerical comparison with STF**
>
> **A3:** We thank the reviewer for the suggestion. Since STF was originally proposed for diffusion models and does not support conditional generation, we adapted its formulation to the flow-matching framework and evaluated it on the synthetic GMM learning task described in Appendix E.1.4. In this setting, the model is trained to match a known Gaussian mixture distribution, and performance is measured via the second-order moment of the discrepancy between the predicted velocity field  $v_\theta(x_t, t)$  and the ground-truth velocity field  $v_t(x_t)$, under the marginal distribution  $p_t(x_t)$. We report results at  $t = 0.30$  for reference and will include the full details and additional results in Appendix E.1.4. The comparison is as follows:
>
> | Iterations | 5k | 10k | 15k | 20k |
> | --- | --- | --- | --- | --- |
> | CFM | 4.6790 | 4.4938 | 4.2470 | 4.1262 |
> | STF (n=2048) | 4.6613 | 4.4506 | 4.2565 | 4.1693 |
> | StableVM (n=2048) | 4.5986 | 4.2404 | 4.0748 | 4.0054 |
>
> As shown in the table, STF provides faster initial improvement but plateaus and ultimately yields the worst long-run performance, consistent with the bias issue we discuss in the paper and Q2. In contrast, StableVM provides consistent improvements throughout training and achieves the best performance across all iterations.

---

### Official Review · Reviewer_7N2v · 2025-10-31

**Soundness:** 2
**Presentation:** 3
**Contribution:** 2
**Rating:** 4
**Confidence:** 3

**Summary:**

This paper analyzes variance in Conditional Flow Matching, identifying high-variance and low-variance regimes. It proposes Stable Velocity Matching, an unbiased variance-reduced training objective using multiple reference samples, and Stable Velocity Sampling, a training-free acceleration method exploiting analytical PF-ODE solutions in the low-variance regime. Experiments demonstrate sampling speedup on large pretrained models.

**Strengths:**

1. The observation regarding variance regimes is well-motivated. The validation experiment in Figure 1 effectively demonstrates that the split point ξ shifts toward 1 as dimensionality increases, supporting the analysis.
2. The experimental validation is comprehensive and involves substantial computational resources.

**Weaknesses:**

1. The proposed accelerated sampling method (StableVS) is only compared to a "default sampler". What is the "default sampler" mentioned in Figure 4? It lacks comparison with other advanced samplers such as DPM-Solver++, etc.
2. Based on Figure 3, the convergence speedup from StableVM appears marginal.
3. The StableVS sampler's performance depends critically on the split point $\xi$. The paper lacks a principled method or heuristic for choosing this value, which appears to be a "magic number" hyperparameter.
4. Does the two-regime variance structure hold true in latent space? It lacks a similar experiment as Figure 1. For latent spaces generated by different VAEs, how would one determine the split point $\xi$?

**Questions:**

1. What is the computational cost that Stable Velocity Matching adds compared to the single-sample Monte Carlo estimate? How does this cost scale as the reference batch size $n$ increases?
2. How was the split point $\xi$ chosen?
3. How will the proposed memory bank solve the sparsity problem for text-conditional generation?
4. What are the generation results (FID, IS) for your ImageNet-trained SiT-XL model when using a low NFEs?

---

> ### Author Response · Authors · 2025-11-20
> **Response to Reviewer 7N2v (Part 1/2)**
>
> Thank you for the insightful and constructive comments. We appreciate your advice and address the questions below.
>
> **Q1: What is the "default sampler" mentioned in Figure 4? Comparison with other advanced samplers such as DPM-Solver++ lacks.**
>
> **A1:** The “default sampler” refers to the one used by each model in its official release. Specifically, SD3, SD3.5, and Flux.1-dev adopt the Euler solver implemented as `FlowMatchEulerDiscreteScheduler` in *Diffusers*, while Wan2.2 employs the UniPC solver [1] (`UniPCMultistepScheduler`) following its *Diffusers* implementation. The detailed hyperparameters for each solver are provided in Appendix E.2.1.
>
> Our proposed sampler is designed primarily for the **low-variance regime** and is compatible with existing solvers in the **high-variance regime**, enabling hybrid sampling. In practice, users may employ Euler, Heun, DPM, or UniPC solvers during the high-variance stage and seamlessly transition to StableVS in the low-variance stage.
>
> Regarding DPM-Solver and DPM-Solver++, we highlight the conceptual connection to our method as also discussed in the related works section. Both DPM solvers exploit the semi-linear structure of PF-ODE dynamics to avoid explicit step-by-step integration. In contrast, **StableVS leverages the property that the conditional velocity closely approximates the true velocity in the low-variance regime**, allowing us to reduce the PF-ODE to a linear ODE. This property extends naturally to SDEs as well. From this perspective, StableVS shares conceptual insights with DPM solvers but offers a more principled simplification based on the variance-dependent behavior of the velocity field.
>
> Below we also report DPM-Solver++ for flow-matching [2–3] and their combination with our StableVS on Stable Diffusion 3.5 Large:
>
> | **Model** | NFE | Overall | PNSR | SSIM | LPIPS |
> | --- | --- | --- | --- | --- | --- |
> | DPM Solver++ | 20 | 0.715 |  |  |  |
> | DPM Solver++ | 15 | **0.717** | 18.245 | 0.796 | 0.268 |
> | DPM Solver++ with StableVS  | 15 | 0.711 | **27.941** | **0.931** | **0.086** |
>
> These results show that with fewer NFEs, **DPM-Solver++ with StableVS matches with the 20-step DPM-Solver++**, supporting both the correctness of our low-variance-regime assumption and the compatibility of StableVS with advanced ODE solvers.
>
> **Q2: The convergence speedup from StableVM appears marginal.**
>
> **A2:** The seemingly modest speedup is primarily due to our use of *classifier-free guidance* (CFG). Prior works such as SiT[4] and REPA[5] report convergence curves *without* CFG, under which FID values are substantially larger and gains are more visible. CFG significantly reduces the performance gap near convergence, making improvements appear smaller.
>
> To make this effect explicit, we compare SiT-XL/2 and our StableVM-enhanced model at 2M iterations under both evaluation settings:
>
> | **Model** | **CFG FID ↓** | **non-CFG FID ↓** |
> | --- | --- | --- |
> | SiT-XL/2 | 2.05 | 10.62 |
> | + StableVM (bank=1024) | **1.94** | **8.87** |
>
> Under the non-CFG setting—consistent with SiT and REPA—the convergence improvement is much more pronounced (10.62 → 8.87). Even with CFG, StableVM still achieves a measurable gain (2.05 → 1.94). We will update Figure 3 accordingly in the revised version to reflect this.
>
> **Q3: What is the computational cost that Stable Velocity Matching? How does this cost scale as the reference batch size $n$ increases?**
>
> **A3:** Stable Velocity Matching introduces additional computation and memory overhead that scales linearly with the reference batch size, i.e. $\mathcal{O}(n)$. Specifically, The self-normalized target requires computing a weighted average over the reference batch, and for class-conditioned generation, the reference batch stores $n$ latent representations per class. For example, in an ImageNet $256\times256$ generation task, we will store 256 latents for 1000 classes in fp16 precision, which requires roughly 2 GB additional memory, and the computation adds only minor latency, which is negligible compared to model’s forward and backward pass.
>
> **Q4: How will the proposed memory bank solve the sparsity problem for text-conditional generation?**
>
> **A4:** Unfortunately, the proposed memory bank cannot fully resolve the sparsity issue in text-conditional generation, as current mainstream datasets rarely contain enough effective samples for the exact same prompt. A possible direction for future work is to incorporate images associated with *similar* prompts, weighted by semantic similarity metrics such as CLIP scores, to enrich the reference set.

---

> ### Author Response · Authors · 2025-11-20
> **Response to Reviewer 7N2v (Part 2/2)**
>
> **Q5: Does the two-regime variance structure hold true in latent space? How was the split point $\xi$ chosen?**
>
> **A5:** Yes, the two-regime variance structure also holds in latent space. Intuitively, as long as latent representations remain well separated across samples, the diffusion process requires a non-trivial amount of time before these latents mix into an indistinguishable Gaussian distribution. As illustrated in Figure 2, for small timesteps $t$ (low-variance regime), the latents have not yet mixed, and noise-perturbed latents typically originate from a single underlying latent. For large $t$ (high-variance regime), the latents are well mixed, leading to substantially higher variance.
>
> To further verify this behaviour, we compute the variance curve for ImageNet latents (size $32\times 32\times 4$). A subset of the results is shown below:
>
> | Timestep (t) | 0.73 | 0.78 | 0.84 | 0.89 | 0.94 | 0.99 |
> | --- | --- | --- | --- | --- | --- | --- |
> | Variance / $\sqrt{\text{dim}}$ | $9.80\times10^{-6}$ | $5.72\times10^{-5}$ | $1.06\times10^{-4}$ | $1.26$ | $33.03$ | $47.13$ |
>
> These measurements confirm that the two-regime structure persists in ImageNet latent space. They also reveal a clear *shifting phenomenon*: the split point $\xi$ moves closer to 1 as the dimensionality of the latent space increases, compared to CIFAR-10 in Figure 1.
>
> Although the exact split point depends on the (unknown) data distribution, it can be inferred through this dimensionality-dependent shift. Higher-dimensional data distributions—such as those in high-resolution T2I or T2V models—are expected to have a split point closer to 1. In practice, this implies  $\xi$ should be greater than the split point of CIFAR-10, i.e. $\xi > 0.7$. Our ablation in Table 6 (Appendix E.2.2) supports this inference: setting ($\xi = 0.7$) yields generation quality identical to the original 50-step sampler, indicating that the true split point for ImageNet latents is indeed larger than that of CIFAR-10.
>
> **Q6: What are the generation results (FID, IS) for your ImageNet-trained SiT-XL model when using a low NFEs?**
>
> **A6:** The quantitative results for SiT-XL with 50 and 250 NFEs are summarized below:
>
> | Model | FID ↓ | IS ↑ |
> | --- | --- | --- |
> | SiT-XL/2 (NFEs=250) | 2.05 | 255.4 |
> | + StableVM (bank=1024) | **1.94** | **265.3** |
> | SiT-XL/2 (NFEs=50) | 2.42 | 245.8 |
> | + StableVM (bank=1024) | **2.16** | **253.3** |
>
> As shown in the table, StableVM consistently improves both FID and IS across different sampling budgets. Notably, the performance gain becomes more pronounced at lower NFEs, suggesting that StableVM accelerates and improve convergence.
>
> **References:**
>
> [1] Zhao, Wenliang, et al. “UniPC: A unified predictor–corrector framework for fast sampling of diffusion models.” *Advances in Neural Information Processing Systems* 36 (2023): 49842–49869.
>
> [2] Lu, Cheng, et al. "Dpm-solver++: Fast solver for guided sampling of diffusion probabilistic models." *Machine Intelligence Research* (2025): 1-22.
>
> [3] Xie, Enze, et al. "Sana: Efficient high-resolution image synthesis with linear diffusion transformers." *arXiv preprint arXiv:2410.10629* (2024).
>
> [4] Ma, Nanye, et al. "Sit: Exploring flow and diffusion-based generative models with scalable interpolant transformers." *European Conference on Computer Vision*. Cham: Springer Nature Switzerland, 2024.
>
> [5] Yu, Sihyun, et al. "Representation alignment for generation: Training diffusion transformers is easier than you think." *arXiv preprint arXiv:2410.06940* (2024).

---

### Official Review · Reviewer_kKja · 2025-11-01

**Soundness:** 2
**Presentation:** 3
**Contribution:** 2
**Rating:** 2
**Confidence:** 5

**Summary:**

This paper proposes Stable Velocity, a framework to improve both training stability and sampling efficiency in flow-matching and stochastic-interpolant generative models. The key insight is that Conditional Flow Matching (CFM) suffers from high variance in its conditional velocity estimates, leading to slow convergence and unstable training.

**Strengths:**

This paper clearly articulates how CFM variance affects optimization and links it to training and inference regimes.

**Weaknesses:**

1. There have been several recent works that significantly enhance training and inference efficiency at scale, such as MeanFlow [2] and REPA (Representation Alignment for Generation) [1]. Compared to these approaches—which demonstrate substantial speedups and scalability gains—the improvements reported in this paper appear relatively minor. Consequently, the overall contribution of this work appears incremental relative to existing approaches.

[1] REPRESENTATION ALIGNMENT FOR GENERATION: TRAINING DIFFUSION TRANSFORMERS IS EASIER THAN YOU THINK
[2] Mean Flows for One-step Generative Modeling

2. Multiple papers have proposed importance sampling for times for diffusion models for reducing the variance of the diffusion ELBO, empirically showing improved results:
Huang, Chin-Wei, Jae Hyun Lim, and Aaron C. Courville. "A variational perspective on diffusion-based generative models and score matching." Advances in Neural Information Processing Systems 34 (2021): 22863-22876.
Song, Yang, et al. "Maximum likelihood training of score-based diffusion models." Advances in neural information processing systems 34 (2021): 1415-1428.
Variance reduction techniques that make use of control variates rather than importance sampling are not discussed either
Jeha, Paul, et al. "Variance reduction of diffusion model's gradients with Taylor approximation-based control variate." arXiv preprint arXiv:2408.12270 (2024).
A comparison to some of these methods would make the paper stronger.

3. The best FID value for SiT-XL/2 reported in prior work is 2.06, yet Table 1 in this paper only presents FID values at selected training iterations without showing the final or best-achieved score. To fairly assess the improvement brought by the proposed method, the authors should include the best FID attained during training and compare it directly with the SiT-XL/2 benchmark result.

**Questions:**

See weakness

---

> ### Author Response · Authors · 2025-11-20
> **Response to Reviewer kKja**
>
> Thank you for the insightful and constructive comments. We appreciate your advice and address the questions below.
>
> **Q1: Incremental contribution of this work compared to existing recent approaches like MeanFlow and REPA.**
>
> **A1:** We thank the reviewer for bringing up recent approaches such as REPA and MeanFlow, which we already discussed in the related works. Our method is complementary to both and differs in motivation, formulation, and implications.
>
> REPA aligns representations through cosine-similarity–based objectives, whereas our StableVM focuses on **target-side variance reduction** by constructing a self-normalized weighted regression target under a composite conditional distribution. This directly alters the learning signal itself: StableVM controls the variance of the regression target across timesteps, which is an aspect not addressed by REPA.
>
> StableVM is therefore **orthogonal** to REPA. In fact, our target construction can be combined with REPA’s representation alignment by substituting velocities with normalised features, demonstrating methodological compatibility without modifying our core objective.
>
> MeanFlow introduces a new ground-truth field representing the average velocity in contrast to the instantaneous velocity typically modelled in Flow Matching. Our StableVS formulation provides a **new theoretical connection**: under a linear interpolant, the PF-ODE trajectory in low-variance regions collapses to a straight path where the average velocity equals the instantaneous velocity. This observation is not used to train the model but enables an analytical simplification during inference—supporting accelerated sampling without additional finetuning or training cost.
>
> **Q2: Papers that proposed importance sampling for times to reduce the variance of the diffusion ELBO like Huang et al. (2021) and Song et al. (2021) and papers that make use of control variates like Jeha, Paul, et al are not discussed.**
>
> **A2:** We thank the reviewer for pointing out related variance-reduction techniques. Prior works such as Huang et al. (2021) and Song et al. (2021) propose *time-based importance sampling* to stabilize the diffusion ELBO/score-MLE objectives, whose training signals exhibit strong **time-dependent variance**, especially near $t\rightarrow 0$ where the score becomes ill-conditioned and sharply peaked. Jeha et al. (2024) further introduce a *control-variate* formulation that analytically cancels gradient fluctuations specific to the diffusion ELBO structure.
>
> However, these techniques do not directly transfer to our setting. StableVM is formulated under **flow matching with stochastic interpolants (SI)**, where the target is the **conditional instantaneous velocity** ($v=\varepsilon - x_0$). Unlike diffusion scores, this quantity has **bounded variance** and lacks the pathological small-$t$ behavior that motivates time-importance sampling. Likewise, the Taylor-expansion control variate of Jeha et al. is derived for the **diffusion score-matching objective** and does not apply to the flow-matching loss; any analogous control variate for FM would be orthogonal to—and compatible with—our method rather than a substitute.
>
> Instead, StableVM targets a fundamentally different and FM-specific source of variance—**the variability across individual reference pairs $(x_0,\varepsilon)$.** Our method reduces this variance by constructing a self-normalized, reference-aggregated regression target and further eliminating bias via a composite conditional formulation. These mechanisms provide variance reduction precisely where diffusion-based techniques are ineffective. We will add explicit discussion of these works in related works of the revised manuscript.
>
> **Q3: The authors should include the best FID attained during training and compare it directly with the SiT-XL/2 benchmark result.**
>
> **A3:** We thank the reviewer for the suggestion. Due to limited computational resources, both the baseline and our method were trained for 2M iterations. The best FID and Inception Score (IS) for SiT-XL/2 and our StableVM variants are reported in Appendix E.1.3. For completeness, we also include results for StableVM with an increased bank capacity of $K=1024$. The key results are summarized below:
>
> | Model | FID ↓ | IS ↑ |
> | --- | --- | --- |
> | SiT-XL/2 | 2.05 | 255.4 |
> | + StableVM (bank=256) | 2.03 | 260.1 |
> | + StableVM (bank=1024) | **1.94** | **265.3** |
>
> As shown, StableVM consistently improves both FID and IS over the SiT-XL/2 baseline, achieving a best FID of 1.94 and an IS of 265.3. These results demonstrate clear and stable quantitative gains over the standard CFM loss.

---

### Meta-Review · Area_Chair_hDLg · 2026-01-01

**Summary:**

This paper introduced a framework named Stable Velocity. The framework improves up on flow matching stability and sampling efficiency by analyzing conditional flow matching (CFM) variance and identifying two regimes. StableVM (high variance regime) is an unbiased, variance-reduced objective that improves convergence, and StableVS (low-variance) is a training-free method for accelerated sampling, enabling upto 2x faster, high-quality samples.

**Reviewer Concerns:**

The main concerns focused on limited novelty: StableVM was related to the STF and StableVS to simple DDIM samples. Reviewers also noted marginal convergence speedups and a lack of comparison with SOTA solvers like DPM-Solver++. A non-principled method for selecting the regime split-point was another weakness shared by reviewers. The authors responded that StableVM is a novel and unbiased target and StableVS is a consequence of the resulting variance analysis that's compatible with DPM-Solver++(advanced solvers).

**Reviewer Scores:**

The paper mostly received 3x reject and 1x marginally-below-acceptance-threshold.

---

### Decision · Program_Chairs · 2026-01-26

Reject